# ClearSR: Latent Low-Resolution Image Embeddings Help Diffusion-Based Real-World Super Resolution Models See Clearer

## Abstract

We present ClearSR, a new method that can better take advantage of latent low-resolution image (LR) embeddings for diffusion-based real-world image super-resolution (Real-ISR). Previous Real-ISR models mostly focus on how to activate more generative priors of text-to-image diffusion models to make the output high-resolution (HR) images look better. However, since these methods rely too much on the generative priors, the content of the output images is often inconsistent with the input LR ones. To mitigate the above issue, in this work, we explore using latent LR embeddings to constrain the control signals from ControlNet, and extract LR information at both detail and structure levels. We show that the proper use of latent LR embeddings can produce higher-quality control signals, which enables the super-resolution results to be more consistent with the LR image and leads to clearer visual results. In addition, we also show that latent LR embeddings can be used to control the inference stage, allowing for the improvement of fidelity and generation ability simultaneously. Experiments demonstrate that our model can achieve better performance across multiple metrics on several test sets and generate more consistent SR results with LR images than existing methods. Our code will be made publicly available.

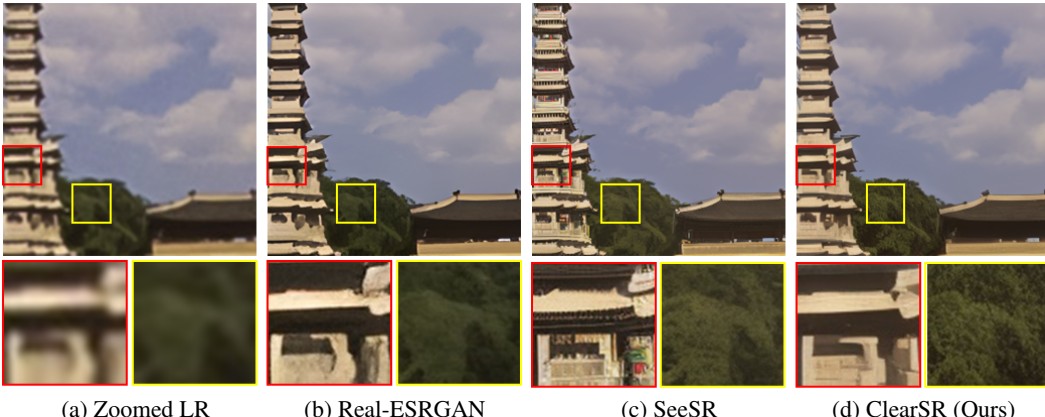

| (a) Zoomed LR | (b) Real-ESRGAN | (c) SeeSR | (d) ClearSR (Ours) |

Figure 1: Visual comparisons with recent state-of-the-art Real-ISR methods. Real-ESRGAN (Wang et al., 2021) results in a lack of generated details. SeeSR (Wu et al., 2024) uses semantic information to activate more generative priors of the SD model but results in **inconsistent** content with the LR image. Our results can properly generate details and have better visual effects.

## 1 Introduction

Real-world Image Super-Resolution (Real-ISR) aims to restore a high-resolution (HR) image from its low-resolution (LR) version in real-world scenarios. Unlike traditional Image Super-Resolution (ISR), Real-ISR requires modeling complex degradations in the real world, which further tests the

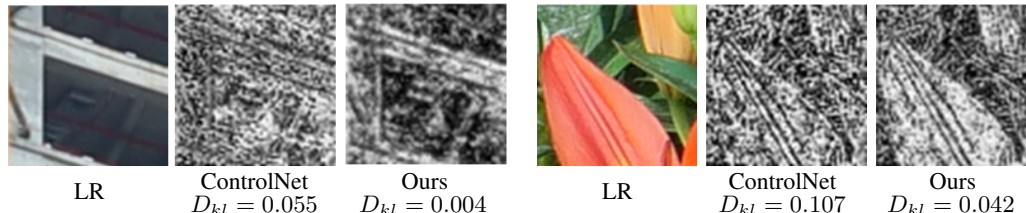



| LR | ControlNet
$D_{kl} = 0.055$ | Ours
$D_{kl} = 0.004$ | LR | ControlNet
$D_{kl} = 0.107$ | Ours
$D_{kl} = 0.042$ |



Figure 2: Analysis of the role of the latent LR embeddings constraint. $D_{kl}$ represents the KL divergence between the control signals and latent LR embeddings. We visualize the control signals with PCA (Parmar et al., 2024). One can observe that the control signals of ControlNet have higher $D_{kl}$ and cannot preserve the LR information well. However, our results have lower $D_{kl}$ and have sharper outlines, indicating that our model can extract LR information better.

models' capability of generating image details. Some researchers (Wang et al., 2021; Chen et al., 2022; Zhang et al., 2021) have used stacked convolutional blocks or transformer-based blocks to build models, or GANs to help generate details, achieving remarkable results. However, because of insufficient generative capability, these models are limited in generating fine details.

Recently, Diffusion Models (DMs) have achieved notable performance in various tasks. Specifically, the pre-trained text-to-image (T2I) models (Saharia et al., 2022; Rombach et al., 2022), such as Stable Diffusion (SD), have a gift in powerful generative priors, which can help generate details needed for Real-ISR. Since then, many SD-based Real-ISR works (Wang et al., 2024a; Lin et al., 2023; Yang et al., 2023; Wu et al., 2024; Yu et al., 2024a; Sun et al., 2024) have emerged. However, pre-trained SD models are originally designed for image generation and directly use them for Real-ISR as done in previous work (Wang et al., 2024a) may lead to super-resolution results with inconsistent content with the input LR images because of the generative priors. Therefore, how to take advantage of the generation capability of SD models in a proper way to avoid the generation of inconsistent content has become a challenge on this topic.

A common approach to mitigate the above issue in previous work (Lin et al., 2023) is to use diffusion adapters, such as ControlNet (Zhang et al., 2023), to process the LR image. For instance, Yang et al. (2023) introduced additional cross-attention layers to integrate the control signals produced by ControlNet into the UNet, demonstrating better consistency between the output and the input LR images. However, this method mainly focuses on the utilization of the control signals but does not consider the way of constructing high-quality control signals. In addition, some works (Wu et al., 2024; Sun et al., 2024) extract semantic information from LR images to activate more generative abilities of SD models, but semantic information is relatively coarse and difficult to provide pixel-wise control.

As mentioned above, diffusion-based Real-ISR models often generate too much content that is inconsistent with the input image as shown in Figure 1. We argue that a key issue that leads to this problem is the inefficient use of the LR image information. As shown in Figure 2, the visualization results show that the control signals from ControlNet cannot preserve the LR information well at both the structure and detail levels.

We tackle this by using the latent LR embeddings to constrain ControlNet as we found that the latent LR embeddings from the pre-trained VAE encoder preserve rich LR information that is beneficial for controlling generative priors from SD. Compared to semantic information, latent LR embeddings can provide more precise controls, an effective way to mitigate the issue of inconsistent SR results. We take advantage of the latent LR embeddings by designing two new modules, called Detail Preserving Module (DPM) and Global Structure Preserving Module (GSPM), which aim to embed the latent LR embeddings through window-based cross-attention into different layers of ControlNet to enhance details, and preserve LR structural information, respectively.

Moreover, we show that the use of latent LR embeddings in the inference stage is also able to address the limitation of previous methods that could only enhance fidelity while not improving generative capability. We achieve this by introducing the Latent Space Adjustment (LSA) strategy. This strategy uses latent LR embeddings to adjust the latent space at both earlier and later timesteps, allowing for a wide range of adjustments to the super-resolution results (over 2dB in PSNR and 0.1 in MANIQA).

With appropriate settings, both the fidelity and generative capability of the model can be enhanced simultaneously.

Extensive experiments demonstrate that our ClearSR has superior generation capabilities and can produce more accurate super-resolution results. As shown in Figure 1, one can observe that our results can properly generate details and have better visual effects, which proves the effectiveness of our method. Our contributions can be summarized as follows:

- We propose ClearSR, a novel method that can improve the utilization efficiency of LR information. The cores are the DPM and GSPM that can constrain the ControlNet in latent space and extract more LR information at both the detail and structure levels.
- We show that the latent LR embeddings can be used to adjust the latent space during the inference stage, which brings improvement of the fidelity and generation ability simultaneously.
- Our proposed ClearSR outperforms previous models on multiple metrics on different test sets. The super-resolution results generated by ClearSR contain rich generated details and meanwhile show better consistency with LR images.

## 2 RELATED WORK

### 2.1 IMAGE SUPER-RESOLUTION

Image Super-Resolution (ISR) aims to restore a high-resolution (HR) image from its low-resolution (LR) version. Traditional ISR works are usually based on stacked CNN or transformer layers and are learned under a known degradation. Since SRCNN (Dong et al., 2015) introduced CNN into the field of image super-resolution and achieved better results than traditional methods, many excellent works have emerged (Dong et al., 2015; Zhang et al., 2018a; Dai et al., 2019; Niu et al., 2020; Tong et al., 2017; Kim et al., 2016; Zhang et al., 2018b; Shi et al., 2016; Ahn et al., 2018; Lim et al., 2017; Mei et al., 2021; Dong et al., 2016; Li et al., 2020). After that, some researchers applied Swin Transformer (Liu et al., 2021) to the image super-resolution task and achieved impressive success (Liang et al., 2021; Chen et al., 2023a; Zhou et al., 2023; Chen et al., 2023b; Zhang et al., 2022). However, as the degradation is usually simple and known, the application scope of this task is limited. In recent years, attention has shifted toward more practically valuable topics, such as Real-world Image Super-Resolution (Zhang et al., 2021; Liang et al., 2022a; Wang et al., 2021; 2024a; Lin et al., 2023; Wu et al., 2024; Yang et al., 2023; Xie et al., 2023).

### 2.2 REAL-WORLD IMAGE SUPER-RESOLUTION

Real-world Image Super-Resolution (Real-ISR) has become a popular topic in recent years. Compared to traditional ISR, Real-ISR requires modeling complex degradations in the real world, which further tests the generative capabilities of models and offers greater practical values. Many studies have used GANs (Wang et al., 2021; Zhang et al., 2021; Liang et al., 2022b) for Real-ISR tasks due to its excellent detail generation capabilities, demonstrating competitive results (Zhang et al., 2021; Wang et al., 2021; Liang et al., 2022b;a). However, GAN-based methods often produce unnatural artifacts, limiting their applications in Real-ISR tasks. Recently, since the introduction of DDPM (Ho et al., 2020), Diffusion Models (DMs) have secured a significant position in the field of image synthesis. After some exploration (Lu et al., 2022; Kong & Ping, 2021; San-Roman et al., 2021), Rombach et al. (2022) reduced the computational cost of DMs, broadening its application range.

Due to the outstanding success of DMs in various computer vision tasks, some researchers have begun to use them for Real-ISR tasks (Yue et al., 2024; Wang et al., 2024b; Xia et al., 2023), but the generative capabilities of these models are still limited. As the pre-trained text-to-image (T2I) DMs, such as Stable Diffusion (SD) have powerful generative priors, which can help generate details needed for Real-ISR. StableSR (Wang et al., 2024a) has used SD for the first time to conduct Real-ISR tasks and demonstrates impressive detail generation capabilities. However, the overly strong generative ability of the pre-trained T2I models often leads to inconsistent super-resolution results. Therefore, how to effectively utilize LR information becomes a challenge in this topic. DiffBIR (Lin et al., 2023) has used ControlNet (Zhang et al., 2023) to provide appropriate control signals for SD, improving the generation effect of the model. On this basis, PASD (Yang et al., 2023) focuses on the control signals provided by ControlNet, making more efficient use of them. SeeSR (Wu et al., 2024) uses

a reasonable method to extract the semantic signals of the LR image to activate more generative abilities of models.

Our work also focuses on how to utilize LR information better. Unlike previous methods (Yang et al., 2023; Wu et al., 2024), we find PASD (Yang et al., 2023) did not improve the control signals themselves, and the usage of semantic information Wu et al. (2024) is coarse and leads to inconsistent SR results. As a result, we focus on latent LR embeddings provided by the pre-trained VAE encoder and use it to constrain the control signals and extract more LR information at both detail and structure levels.

## 3 METHODOLOGY

### 3.1 OVERALL ARCHITECTURE OF CLEARSR

As mentioned in Sec. 1, the inefficient LR image utilization may lead to inconsistency with the input image. Therefore, our intention is to dig how to better take advantage of the LR information to improve the control signals. Figure 3 shows the overall pipeline of our ClearSR. The cores of our method are two new modules, named Detail Preserving Module (DPM) and Global Structure Preserving Module (GSPM). DPM is based on ControlNet, which aims to preserve essential detailed information from the latent LR embeddings for the control signals. GSPM on the other hand, is a ResBlock-based module without attention layers, allowing it to retain more LR structure information and enhance fidelity (Yu et al., 2024b).

During the training process, the objective of the Diffusion Model (DM) is to learn the probability distribution of the reverse denoising process. Specifically, denote the LoRA finetuned SD UNet, the LoRA finetuned VAE encoder, the pre-trained VAE encoder, and the pre-trained VAE decoder as $\epsilon_{\theta'}$, $E_{\theta'}$, $E_\theta$, and $D_\theta$, respectively. Denote DPM and GSPM as $d_\phi$ and $s_\phi$. For a randomly sampled time step $t$ and a high-quality image $\mathbf{I}_{hq}$, let $\mathbf{I}_{hq}$ pass through $E_\theta$ and perform the noise addition process to obtain $\mathbf{x}_t$. Sending the low-quality image $\mathbf{I}_{lr}$ into $E_{\theta'}$ yields the latent LR embeddings $\mathbf{x}_{lr}$. Then, we can collect the control signals $\mathbf{x}_c = \{\mathbf{c}_1, \mathbf{c}_2, \dots\}$ by inputting $\mathbf{x}_{lr}$ into $d_\phi$ and $s_\phi$ and summing their outputs. Similar to PASD (Yang et al., 2023) and CoSeR (Sun et al., 2024), we let the LR image pass through the CLIP image encoder to obtain the image-level feature $\mathbf{p}$ and replace the null-text prompt in the UNet decoder. The optimization objective can be formulated as:

$$\mathcal{L} = \mathbb{E}_{\mathbf{x}_t, \mathbf{x}_c, \mathbf{p}, t, \epsilon \sim \mathcal{N}} \left[ \|\epsilon - \epsilon_{\theta'}(\mathbf{x}_t, \mathbf{x}_c, \mathbf{p}, t)\|_2^2 \right], \tag{1}$$

where the $\epsilon$ is the added noise.

As mentioned in a previous work (Yu et al., 2024a), the pre-trained VAE encoder is unsuitable for encoding LR images, because it was not trained on LR images. During training, unlike previous works, such as SUPIR (Yu et al., 2024a) and SeeSR (Wu et al., 2024), that introduce a new loss or design a new encoder, we simply add LoRA layers to the pre-trained VAE encoder to tackle this issue. Therefore, there is no need to separately train an encoder or design a new encoder. We also add LoRA layers to the SD UNet decoder to adapt the model to the mixed control signals. Readers may refer to Appendix A for more details.

### 3.2 HIGH-QUALITY CONTROL SIGNAL MODELING

As mentioned above, we intend to produce high-quality control signals by better taking advantage of LR images. We achieve this by adjusting the control signals at both the detail and structure levels, which corresponds to two new modules, called Detail Preserving Module (DPM) and Global Structure Preserving Module (GSPM), respectively. In what follows, we will give their detailed descriptions.

**Detail Preserving Module.** Our DPM aims to constrain ControlNet at the detail level. As ControlNet contains generative priors of SD, the detailed information of latent LR embeddings cannot be preserved well (See Figure 2). As a result, we use window-based cross-attention layers to integrate the latent LR embeddings into different layers of ControlNet. These cross-attention layers are placed after text cross-attention layers. Specifically, let the newly added cross-attention layer be denoted as CA. Given the intermediate feature $\mathbf{x}_d \in \mathbb{R}^{L \times C}$, we let it pass through the linear layer and window partition yields $\mathbf{Q} \in \mathbb{R}^{N \times S^2 \times C}$. Then, let the latent LR embeddings $\mathbf{x}_{lr} \in \mathbb{R}^{l \times c}$ pass through a

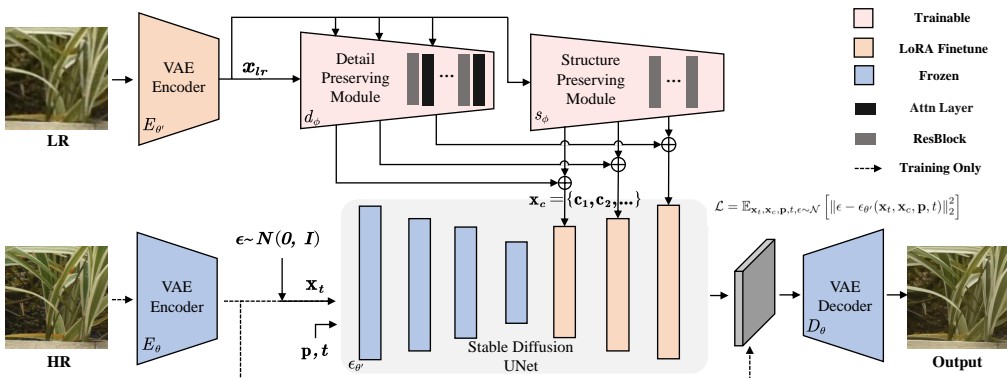

Figure 3: Overview of our ClearSR. Our ClearSR consists of the pre-trained Stable Diffusion (SD), the Detail Preserving Module (DPM), and the Global Structure Preserving Module (GSPM). To produce high-quality control signals, we let the LR image pass through the LoRA finetuned VAE Encoder first to obtain latent LR embeddings $\mathbf{x}_{lr}$. Then, we collect the control signals $\mathbf{x}_c = \{\mathbf{c}_1, \mathbf{c}_2, \dots\}$ by inputting $\mathbf{x}_{lr}$ into the DPM and the GSPM and summing their outputs. We feed the control signals into the decoder of SD UNet to control the HR image generation.

linear layer and window partition, yielding $\mathbf{K} \in \mathbb{R}^{N \times s^2 \times C}$ and $\mathbf{V} \in \mathbb{R}^{N \times s^2 \times C}$, respectively. Here, $N$ is the number of windows, $S$ is the side length of each window of $\mathbf{Q}$, $s$ is the side length of each window of $\mathbf{K}$ and $\mathbf{V}$, $L$ and $C$ are the token number and channel number of $\mathbf{x}_d$, and $l$ as well as $c$ are the token number and channel number of $\mathbf{x}_{lr}$. The formulation can be written as follows:

$$\mathrm{CA}(\mathbf{Q}, \mathbf{K}, \mathbf{V}) = \mathrm{Softmax}\left(\frac{\mathbf{Q}\mathbf{K}^T}{\sqrt{d_k}} + \mathbf{B}\right)\mathbf{V}, \tag{2}$$

where $\mathbf{B}$ is an aligned relative position embedding and $\sqrt{d_k}$ is a scaling factor as defined in (Dosovitskiy et al., 2020).

**Global Structure Preserving Module.** Our GSPM aims to constrain ControlNet at the structure level. GSPM is an independent module that removes the transformer blocks and only retains the ResBlocks from the ControlNet. Since the attention layer is based on weighted calculations between features, it may ignore the original spatial structure. Thus, excluding the attention layers can preserve the structural information (Yu et al., 2024b) that helps generate a consistent HR image with the input LR one. GSPM can present multi-scale control signals consistent in shape with DPM. We sum up the two to form the final control signals $x_c = \{c_1, c_2, \dots\}$.

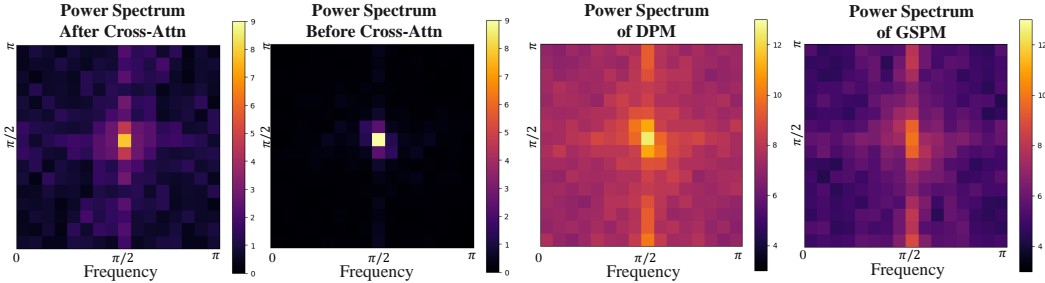

Figure 4: Power spectrum visualization of the intermediate features. The two images on the left show that the cross-attention layer can increase high-frequency information, and the two images on the right show that DPM contains more high-frequency information than GSPM. More results are shown in Appendix C.

**Analysis.** We demonstrate the effectiveness of adding constraints to ControlNet first. As shown in Figure 2, we evaluate the deviation between the control signal and the LR information by calculating the KL divergence $D_{kl}$ between the control signals and the latent LR embeddings. Compared to the

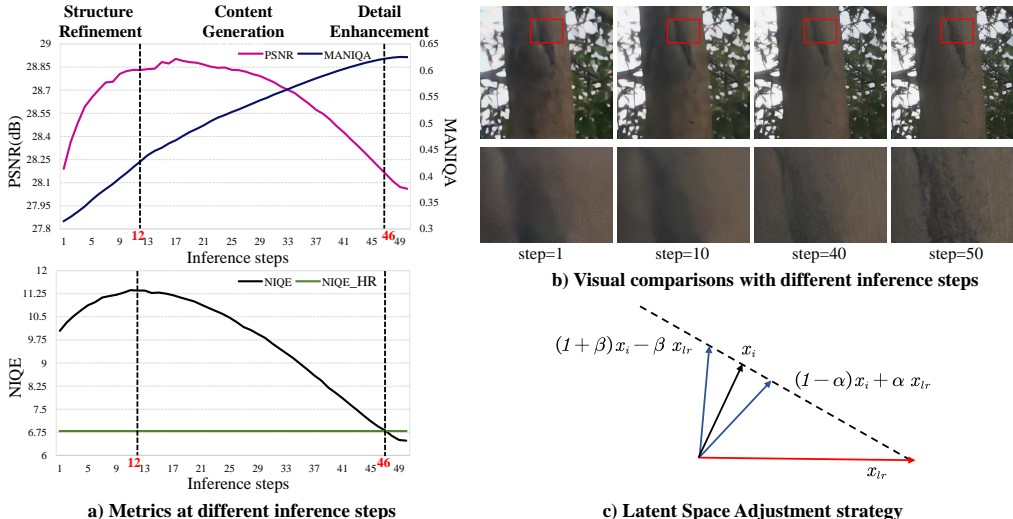

Figure 5: Overview of our Latent Space Adjustment strategy. a) shows the average PSNR, MANIQA, and NIQE curves of the DRealSR test set. b) shows the images at different steps. c) demonstrates our Latent Space Adjustment strategy.

model that only uses ControlNet (also trained), the control signal output of our model exhibits a lower $D_{kl}$, indicating that latent LR embeddings successfully constrain ControlNet. Furthermore, we visualized the control signal using PCA (Parmar et al., 2024), which reveals that our control signal maintains the LR information effectively, demonstrating that our method can make better use of the LR information (See Appendix B for more discussions).

Next, we briefly analyze why our DPM and GSPM help. In Figure 4, we use the power spectrum of intermediate features to validate the effectiveness of our DPM and GSPM. The two images on the left show the power spectrum of features in the DPM before and after passing through one cross-attention layer. We can see that after the cross-attention layer, the intermediate features contain more high-frequency components, indicating that more detailed information has been extracted, which aligns with our design intent. The two images on the right show the power spectrum of the control signals from DPM and GSPM. It can be seen that the output from DPM contains more high-frequency information which is helpful for reconstructing details while GSPM mainly contains low-frequency information which preserves structural information.

### 3.3 LATENT SPACE ADJUSTMENT STRATEGY IN INFERENCE STAGE

Previous work has pointed out that adding additional LR information during the inference stage can help improve fidelity (Yu et al., 2024a; Wu et al., 2024). However, the improvement in fidelity comes at the expense of reducing the generative capability of the models. This type of unidirectional adjustment strategy has a negative impact on the image details after super-resolution, affecting the visual effect. Unlike previous works, we propose the Latent Space Adjustment (LSA) strategy, which can improve either fidelity or generation. Moreover, our strategy can improve the fidelity and generation simultaneously through appropriate settings.

As shown in Figure 5(a), during the inference stage, the PSNR score increases first and then decreases as the number of steps increases. This is because the model performs structural refinement in the early steps while generation in the later steps (Sun et al., 2023). For the middle steps of the inference stage, the model focuses on content generation. As shown in Figure 5(b), at around $40_{th}$ step, the model can already determine most of the information in the image, but it is difficult to generate realistic textures. This indicates that detail enhancement is mainly in the last few steps. This motivates us to divide the whole inference stage into three parts: structure refinement, content generation, and detail enhancement.

Based on the analysis above, we propose the Latent Space Adjustment (LSA) strategy. We notice that an inherent property of LR images is that it mainly contains structural information and has less details compared to HR images. We take advantage of this property and move the output of each inference step away from the latent LR embeddings in the latent space in the later steps of the inference stage so that the model can focus more on generating details (Lin et al., 2023). In contrast, in the early steps, we let the output close to the latent LR embeddings, similar to previous work, to enhance the fidelity. As shown in Figure 5(a), we statistically select the highest point of the NIQE curve and the point where NIQE starts to fall below the HR image to split the inference stage into three parts. The LR adjustments in the structure refinement and detail enhancement are referred to as Early-step LR Adjustment (ELA) and Later-step LR Adjustment (LLA), respectively. We use two factors $\alpha$ and $\beta$ to determine the control level. Figure 5(c) shows our LSA strategy. The $x_i$ is the predicted latent embeddings of $i$-th step. The formulation can be written as follows:

$$\text{ELA}(x_i) = (1 - \alpha)x_i + \alpha x_{lr}, \tag{3}$$

$$\text{LLA}(x_i) = (1 + \beta)x_i - \beta x_{lr}. \tag{4}$$

The experimental results show that using ELA can improve the fidelity of the model, and using LLA can improve the generation, solving the problem that previous methods (Wu et al., 2024; Yu et al., 2024a) can only adjust in one direction. Moreover, the fidelity and generation of the model can be improved simultaneously through appropriate $\alpha$ and $\beta$ settings. (See Section 4.3 for more discussions.)

Table 1: Quantitative comparison of our ClearSR with recent state-of-the-art **Real-ISR** methods on five benchmark datasets. The best performance is marked in **red** and the second best is marked in **blue**. We compare ClearSR* with GAN-based and Diffusion-based methods (no generative priors), and ClearSR with SD-based methods. ClearSR* has the same structure as ClearSR, but with improved fidelity by modifying the LSA settings.

| Datasets | Method | PSNR↑ | SSIM↑ | LPIPS↓ | NIQE↓ | MUSIQ↑ | MANIQA↑ | CLIPIQA↑ |
|---|---|---|---|---|---|---|---|---|
| DRealSR | **Real-ESRGAN** (Wang et al., 2021) | **28.64** | **0.8053** | **0.2847** | **6.6928** | 54.18 | 0.4907 | 0.4422 |
| | **LDL** (Liang et al., 2022b) | 28.21 | **0.8126** | **0.2815** | 7.1298 | 53.85 | **0.4914** | 0.4310 |
| | **ResShift** (Yue et al., 2024) | 28.46 | 0.7673 | 0.4006 | 8.1249 | 50.60 | 0.4586 | 0.5342 |
| | **SinSR** (Wang et al., 2024b) | 28.36 | 0.7515 | 0.3665 | 6.9907 | **55.33** | 0.4884 | **0.6383** |
| | **ClearSR*(ours)** | **29.00** | 0.7781 | 0.3281 | **6.9796** | 62.74 | **0.5878** | **0.6585** |
| | **StableSR** (Wang et al., 2024a) | 28.03 | 0.7536 | **0.3284** | 6.5239 | 58.51 | 0.5601 | 0.6356 |
| | **PASD** (Yang et al., 2023) | 27.36 | 0.7073 | 0.3760 | **5.5474** | 64.87 | **0.6169** | **0.6808** |
| | **DiffBIR** (Lin et al., 2023) | 26.71 | 0.6571 | 0.4557 | 6.3124 | 61.07 | 0.5930 | 0.6395 |
| | **SeeSR** (Wu et al., 2024) | **28.17** | **0.7691** | **0.3189** | 6.3967 | **64.93** | 0.6042 | 0.6804 |
| | **ClearSR(ours)** | **28.22** | **0.7538** | 0.3473 | **6.0867** | 66.27 | **0.6246** | **0.6976** |
| RealSR | **Real-ESRGAN** (Wang et al., 2021) | 25.69 | **0.7616** | **0.2727** | **5.8295** | 60.18 | **0.5487** | 0.4449 |
| | **LDL** (Liang et al., 2022b) | 25.28 | **0.7567** | **0.2766** | 6.0024 | **60.82** | 0.5485 | 0.4477 |
| | **ResShift** (Yue et al., 2024) | **26.31** | 0.7421 | 0.3460 | 7.2635 | 58.43 | 0.5285 | 0.5444 |
| | **SinSR** (Wang et al., 2024b) | **26.28** | 0.7347 | **0.3188** | 6.2872 | 60.80 | 0.5385 | **0.6122** |
| | **ClearSR*(ours)** | 25.86 | 0.7141 | 0.3148 | **5.6775** | **67.70** | **0.6262** | **0.6560** |
| | **StableSR** (Wang et al., 2024a) | 24.70 | **0.7085** | **0.3018** | 5.9122 | 65.78 | 0.6221 | 0.6178 |
| | **PASD** (Yang et al., 2023) | **25.21** | 0.6798 | 0.3380 | 5.4137 | 68.75 | **0.6487** | **0.6620** |
| | **DiffBIR** (Lin et al., 2023) | 24.75 | 0.6567 | 0.3636 | 5.5346 | 64.98 | 0.6246 | 0.6463 |
| | **SeeSR** (Wu et al., 2024) | 25.18 | **0.7216** | **0.3009** | **5.4081** | **69.77** | 0.6442 | 0.6612 |
| | **ClearSR(ours)** | **25.30** | 0.6911 | 0.3318 | **5.0642** | **69.83** | **0.6499** | **0.6960** |
| DIV2K-Val | **Real-ESRGAN** (Wang et al., 2021) | 24.29 | **0.6371** | **0.3112** | **4.6786** | 61.06 | **0.5501** | 0.5277 |
| | **LDL** (Liang et al., 2022b) | 23.83 | **0.6344** | 0.3256 | **4.8554** | 60.04 | 0.5350 | 0.5180 |
| | **ResShift** (Yue et al., 2024) | **24.65** | 0.6181 | 0.3349 | 6.8212 | 61.09 | 0.5454 | 0.6071 |
| | **SinSR** (Wang et al., 2024b) | **24.41** | 0.6018 | **0.3240** | 6.0159 | **62.82** | 0.5386 | **0.6471** |
| | **ClearSR*(ours)** | 24.23 | 0.5958 | 0.3441 | 5.0315 | **66.90** | **0.6118** | **0.6788** |
| | **StableSR** (Wang et al., 2024a) | 23.26 | 0.5726 | **0.3113** | 4.7581 | 65.92 | 0.6192 | 0.6771 |
| | **PASD** (Yang et al., 2023) | 23.14 | 0.5505 | 0.3571 | **4.3617** | **68.95** | **0.6483** | 0.6788 |
| | **DiffBIR** (Lin et al., 2023) | 23.64 | 0.5647 | 0.3524 | 4.7042 | 65.81 | 0.6210 | 0.6704 |
| | **SeeSR** (Wu et al., 2024) | **23.68** | **0.6043** | **0.3194** | 4.8102 | 68.67 | 0.6240 | **0.6936** |
| | **ClearSR(ours)** | **23.86** | **0.5796** | 0.3493 | **4.6146** | **69.34** | **0.6328** | **0.7040** |

## 4 EXPERIMENTS

### 4.1 EXPERIMENT SETTINGS

Following previous works (Wu et al., 2024; Yang et al., 2023), for training, we train ClearSR on DIV2K (Agustsson & Timofte, 2017), Flickr2K (Timofte et al., 2017), DIV8K (Gu et al., 2019), OST (Wang et al., 2018), and the first 10K face images from FFHQ (Karras et al., 2019). We use the degradation pipeline of Real-ESRGAN (Wang et al., 2021) to obtain LR/HR pairs. For testing, we test ClearSR on DRealSR (Wei et al., 2020), RealSR (Cai et al., 2019), and DIV2K-Val (Agustsson & Timofte, 2017) using the same configurations as (Wu et al., 2024). For evaluation, we employ 7 widely used metrics, including PSNR, SSIM, LPIPS, NIQE, MUSIQ, MANIQA, and CLIPIQA. We use PSNR, SSIM (calculated on the Y channel from the YCbCr space) to evaluate fidelity, LPIPS to evaluate perceptual quality, and NIQE, MUSIQ, MANIQA, and CLIPIQA to evaluate the generation ability of the model.

For implementation details, we use SD 2.1-base as our pre-trained T2I model. We use the Adam optimizer to train ClearSR. The total iteration, batch size, learning rate, inference step are set to 150K, 8, $5 \times 10^{-5}$, and 50, respectively. $\alpha$ and $\beta$ are set to 0.01 and 0.01 for ClearSR and 0.03 and 0.01 for ClearSR*, respectively. We also use the LR embeddings proposed by (Wu et al., 2024) in the inference stage to improve the fidelity. The training process is conducted on $512 \times 512$ resolution with 4 NVIDIA A40 GPUs. For inference, we use a spaced DDPM sampling schedule (Nichol & Dhariwal, 2021).

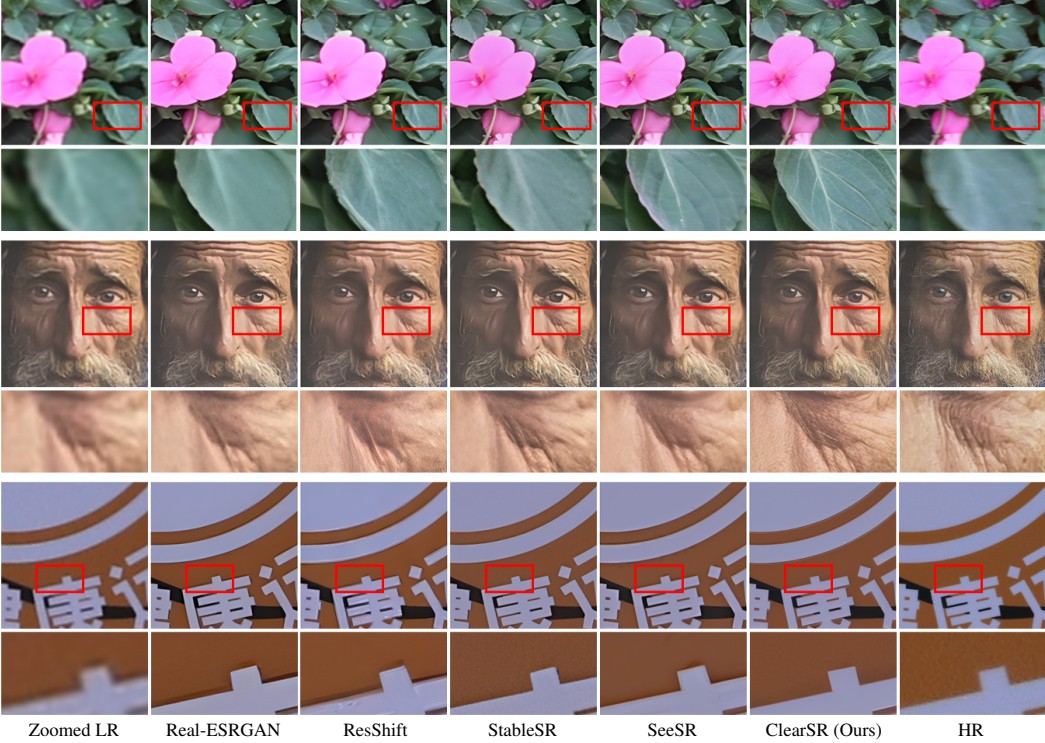

| Zoomed LR | Real-ESRGAN | ResShift | StableSR | SeeSR | ClearSR (Ours) | HR |

Figure 6: Visual comparisons with recent state-of-the-art Real-ISR methods. We can see that the results of our ClearSR have more generated details, and are more consistent with LR images (Zoom in for a better view).

### 4.2 COMPARISONS WITH STATE-OF-THE-ART METHODS

**Quantitative comparisons.** We show the quantitative comparisons between our ClearSR and previous state-of-the-art Real-ISR methods (Wang et al., 2021; Liang et al., 2022b; Yue et al., 2024; Wang et al., 2024b;a; Yang et al., 2023; Lin et al., 2023; Wu et al., 2024) in Table 1. As GAN-based and Diffusion-based (no generative priors) methods (Real-ESRGAN, LDL, ResShift, SinSR) focus more

Table 2: Ablation on model design.

| Model Design | PSNR↑ | SSIM↑ | NIQE↓ | MANIQA↑ |
|---|---|---|---|---|
| w/o GSPM | 27.57 | 0.7490 | 6.9713 | 0.6241 |
| DPM w/o cross-attn layers | 28.06 | 0.7358 | 6.1502 | 0.6100 |
| DPM w/o window partition | 27.60 | 0.7420 | 6.0362 | 0.6273 |
| full model | 27.93 | 0.7455 | 6.2031 | 0.6219 |

Table 3: Ablation on LoRA layers.

| VAE LoRA | UNet LoRA | PSNR↑ | SSIM↑ | NIQE↓ | MANIQA↑ |
|---|---|---|---|---|---|
| - | - | 28.89 | 0.7978 | 7.6531 | 0.5381 |
| ✓ | - | 27.38 | 0.7296 | 6.2722 | 0.6374 |
| - | ✓ | 28.87 | 0.7920 | 7.4344 | 0.5612 |
| ✓ | ✓ | 27.93 | 0.7455 | 6.2031 | 0.6219 |

on fidelity, while SD-based methods focus more on generation, we show the standard ClearSR to compare with GAN-based and Diffusion-based (no generative priors) methods, and compare another version of our ClearSR represented as ClearSR* with modified LSA settings with SD-based methods. As shown in Table 1, it can be seen that our method has advantages on all four generation metrics (NIQE, MUSIQ, MANIQA, CLIPIQA) while maintaining high fidelity (PSNR, SSIM).

**Visual comparisons.** We show the visual comparisons between our ClearSR and previous state-of-the-art Real-ISR methods (Wang et al., 2021; Yue et al., 2024; Wang et al., 2024a; Wu et al., 2024) in the Figure 6. In the first picture, our ClearSR can generate more realistic leaf vein textures, and in the second picture, our ClearSR can generate more realistic facial details, demonstrating the superior generative capability of our ClearSR. Besides, in the third picture, one can observe that our results are clearer than previous methods, proving the effectiveness of our method.

### 4.3 ABLATION ANALYSIS

In this subsection, we conduct extensive experiments to show the effectiveness of our method. For ablation study, the total iteration, batch size, and learning rate are set to 50K, 8, and $1 \times 10^{-4}$, respectively. $\alpha$ and $\beta$ are set to 0.01 and 0.01. We use DRealSR (Wei et al., 2020) for testing and PSNR, SSIM, NIQE, MANIQA metrics for evaluation.

**Effectiveness of Model Design.** We first conduct the ablation study on our model design. As shown in Table 2, we compare the full model with several modified versions. As discussed above, GSPM preserves the structural information and DPM preserves the detailed information. We can see the model without GSPM results in a drop in PSNR, which means a loss of fidelity. Moreover, removing the extra cross-attention layers in DPM leads to a drop in MANIQA, which means a loss for generation. We also testing the window partition strategy. We can see that not using the window partition strategy in DPM weakens the fidelity (PSNR and SSIM metrics). These results prove the effectiveness of our model design.

**Effectiveness of LoRA.** Next, we demonstrate the effectiveness of LR information adaptation. As shown in Table 3, the model without VAE LoRA layers achieves a very high PSNR and SSIM, but its generative capability decreases significantly. This is because the pre-trained VAE encoder cannot correctly map the LR image to the latent space. The model with VAE LoRA layers and without UNet LoRA layers exhibits higher MANIQA but lower fidelity (PSNR and SSIM metrics). This may be due to the fact that the UNet fails to adapt to the output of the mixed control signals, leading to the misapplication of the provided structural information in generating details.

Table 4: Ablation on the Latent Space Adjustment (LSA) strategy. We can see that the LSA strategy can improve fidelity and generation simultaneously.

| ELA $\alpha = 0.01$ | LLA $\beta = 0.01$ | PSNR↑ | SSIM↑ | NIQE↓ | MANIQA↑ |
|---|---|---|---|---|---|
| - | - | 28.11 | 0.7419 | 6.5289 | 0.6226 |
| ✓ | - | 28.44 | 0.7609 | 6.4765 | 0.6172 |
| - | ✓ | 27.85 | 0.7412 | 5.9875 | 0.6360 |
| ✓ | ✓ | 28.22 | 0.7538 | 6.0867 | 0.6246 |

**Effectiveness of latent space adjustment.** We demonstrate the effectiveness of our Latent Space Adjustment (LSA) strategy here. Table 4 shows the effectiveness of LSA on our ClearSR. We can see that with appropriate LSA settings, the fidelity and generation of the model can be improved simultaneously. Furthermore, as shown in Figures 7(a) and (b), increasing $\alpha$ can improve the fidelity (See PSNR score), while increasing $\beta$ can improve the generation ability (See MANIQA score). By using the LSA strategy, the super-resolution results can be adjusted over a wide range (over 2dB in

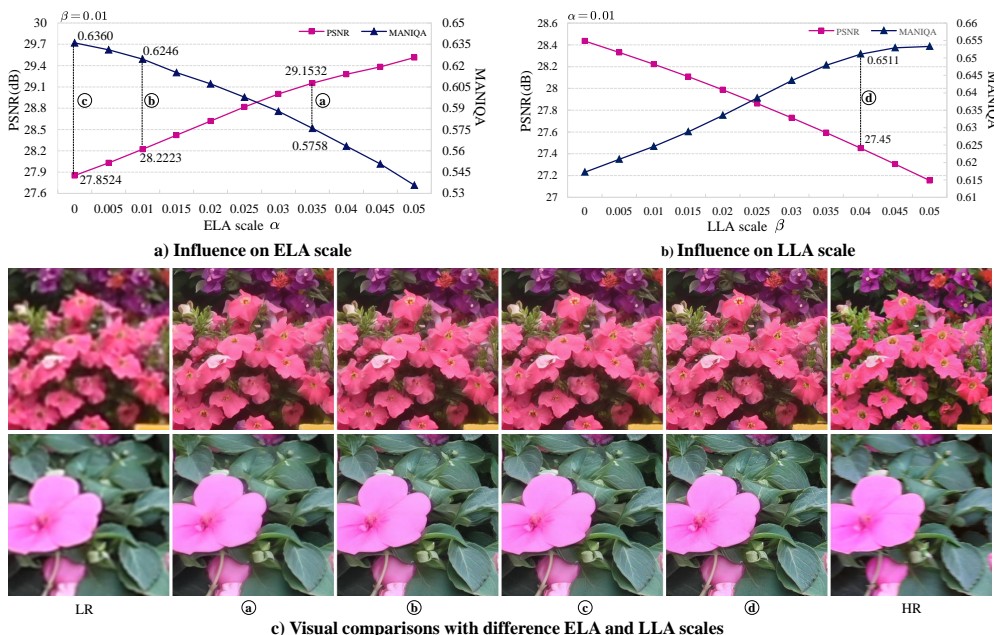

c) Visual comparisons with difference ELA and LLA scales

Figure 7: Impact of the Latent Space Adjustment (LSA) strategy in inference stage. a) and b) show the changes in metrics under different settings. c) shows the results under different LSA settings. One can observe that the super-resolution results can be adjusted over a wide range (over 2dB in PSNR and 0.1 in MANIQA).

PSNR and 0.1 in MANIQA). Figure 7(c) shows the visual comparisons with difference $\alpha$ and $\beta$. We can see that our ClearSR can take into account both fidelity and generation. When the PSNR score is high, our model can still generate meaningful textures instead of overly smooth results.

## 5 CONCLUSIONS

We propose ClearSR, a new method that can better take advantage of latent LR embeddings for diffusion-based Real-ISR tasks. We constrain the ControlNet in latent space through latent LR embeddings, and propose DPM and GSPM to extract LR information at the detail and structure levels. We propose a new LSA strategy in inference stage, which can improve the fidelity and generation ability simultaneously. Extensive experimental results show that the super-resolution results of our ClearSR are more consistent with the LR images and can also generate rich details for better visual effects.

## REPRODUCIBILITY STATEMENT

We provide the experimental settings in Section 4.1. The training and testing sets are all publicly available. The implementation details of the model are also provided. Our code will be made publicly available.

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

## A   MORE DETAILS OF LR INFORMATION ADAPTATION

As mentioned in a previous work (Yu et al., 2024a), the pre-trained VAE encoder is unsuitable for encoding LR images, because it was not trained on LR images. As a result, the pre-trained VAE encoder is unable to map LR images to the correct latent space. Besides, the mixed control signals are also unfamiliar to SD UNet, which is a similar problem. We simply add LoRA layers to the VAE encoder and SD Unet to adapt the LR information to our model. For the VAE encoder, we add LoRA layers to each convolution layer and attention layer, and for UNet, we only use LoRA layers in the attention layer of the UNet decoder. Our method is simple, requires fewer training parameters, and does not need an additional training phase. The LoRA rank is set to 16.

## B   MORE ANALYSIS ON LATENT LR EMBEDDINGS CONSTRAINT

As shown in Figure 8, we calculate the difference in KL divergence Diff on the DRealSR test set. Define the LR image as $\mathbf{I}$, the formula we use to calculate Diff is as follows:

$$\text{Diff}(\mathbf{I}) = D_{kl(ControlNet)} - D_{kl(Ours)}, \tag{5}$$

where $D_{kl(ControlNet)}$ is the KL divergence between the control signal of ControlNet and the latent LR embeddings of $\mathbf{I}$, the $D_{kl(Ours)}$ is the KL divergence between the control signal of our model and the latent LR embeddings of $\mathbf{I}$. One can observe that in Figure 8, the Diff values for most images are greater than 0, proving that our method can effectively reduce $D_{kl}$. This result indicates that our method can effectively use latent LR embeddings to constrain ControlNet in the latent space.

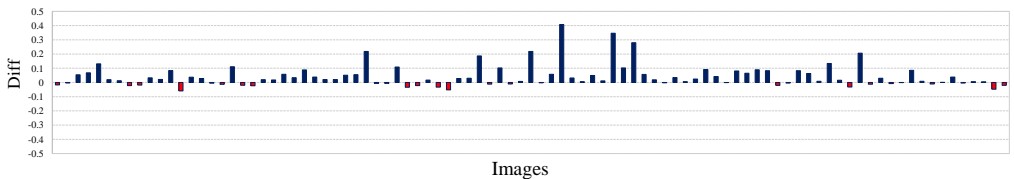

Figure 8: The difference in KL divergence on the DRealSR test set. We can see that our method effectively reduces $D_{kl}$.

## C   MORE POWER SPECTRUM RESULTS

We show more power spectrum results in Figure 9, which further proving the effectiveness of our DPM and GSPM.

## D   MORE VISUAL COMPARISONS

More visual comparisons are shown in Figure 10. It can be seen that in various scenarios, such as buildings, trees, and rocks, our model can produce super-resolution results that are more consistent with the LR image and generate more details.

## E   COMPLEXITY COMPARISONS

In this section, We compare the complexity of our ClearSR with that of recent state-of-the-art SD-based Real-ISR methods (Lin et al., 2023; Yang et al., 2023; Wu et al., 2024), including total parameters, trainable parameters, MACs, inference step, inference time and inference speed. All methods are tested on an A40 GPU. As shown in Table 5, although the additional layers increase the number of parameters and computational cost, we can see that our ClearSR has fewer total parameters, trainable parameters, and MACs compared to SeeSR. For inference speed, since the Diffusers library is optimized for the Classfier-Free Guidance (CFG), we disabled CFG during inference to achieve a fair comparison. Note that DiffBIR originally does not use CFG.

Table 5: Complexity comparisons with recent state-of-the-art SD-based Real-ISR methods.

|  | DiffBIR | PASD | SeeSR | ClearSR |
|---|---|---|---|---|
| Total Param (M) | 1717 | 1900 | 2524 | 2511 |
| Trainable Param (M) | 380 | 625 | 750 | 525 |
| MACs (G) | 24234 | 29125 | 65857 | 52384 |
| Inference Steps | 50 | 20 | 50 | 50 |
| Inference Time (s) | 4.51 | 1.92 | 4.10 | 5.36 |
| Inference Speed (step/s) | 11.09 | 10.41 | 12.21 | 9.33 |

# F    MORE ABLATION ANALYSIS

In this section, the total iteration, batch size, and learning rate are set to 50K, 8, and $5 \times 10^{-5}$, respectively.

**Ablation on the window size.** We conduct the ablation study on the window size of the window-based cross-attention layers in DPM. As shown in Table 6, we can see that increasing the window size leads to a decrease in fidelity, while decreasing the window size reduces the model's generative ability. To balance fidelity and generative ability, we finally chose a window size of 16 for ClearSR.

**Ablation on the LoRA rank.** We also conduct the ablation study on the LoRA rank. Table 7 shows the ablation results for the VAE LoRA rank. As seen, reducing LoRA rank improves fidelity but has a negative impact on generation metrics. On the contrary, increasing LoRA rank has a negative impact on fidelity but improves generation metrics. To balance fidelity and generation, we finally set the VAE LoRA rank to 16. Table 8 shows the ablation results for the UNet LoRA rank. We can see that both smaller LoRA rank and larger LoRA rank improve fidelity but have a negative impact on generation metrics. To balance fidelity and generation, we finally set the UNet LoRA rank to 16.

Table 6: Ablation on the window size.

| Window size | PSNR↑ | SSIM↑ | NIQE↓ | MANIQA↑ |
|---|---|---|---|---|
| 32 | 27.29 | 0.7294 | 6.5364 | 0.6333 |
| 16 | 27.62 | 0.7483 | 6.6334 | 0.6222 |
| 8 | 27.97 | 0.7619 | 6.9919 | 0.6090 |

Table 7: Ablation on the VAE LoRA rank.

| VAE LoRA rank | PSNR↑ | SSIM↑ | MUSIQ↑ | MANIQA↑ |
|---|---|---|---|---|
| 8 | 27.70 | 0.7512 | 66.64 | 0.6221 |
| 16 | 27.62 | 0.7483 | 66.80 | 0.6222 |
| 32 | 27.46 | 0.7346 | 67.06 | 0.6311 |

Table 8: Ablation on the UNet LoRA rank.

| VAE LoRA rank | PSNR↑ | SSIM↑ | MUSIQ↑ | MANIQA↑ |
|---|---|---|---|---|
| 8 | 27.70 | 0.7508 | 66.28 | 0.6165 |
| 16 | 27.62 | 0.7483 | 66.80 | 0.6222 |
| 32 | 27.83 | 0.7533 | 66.35 | 0.6166 |

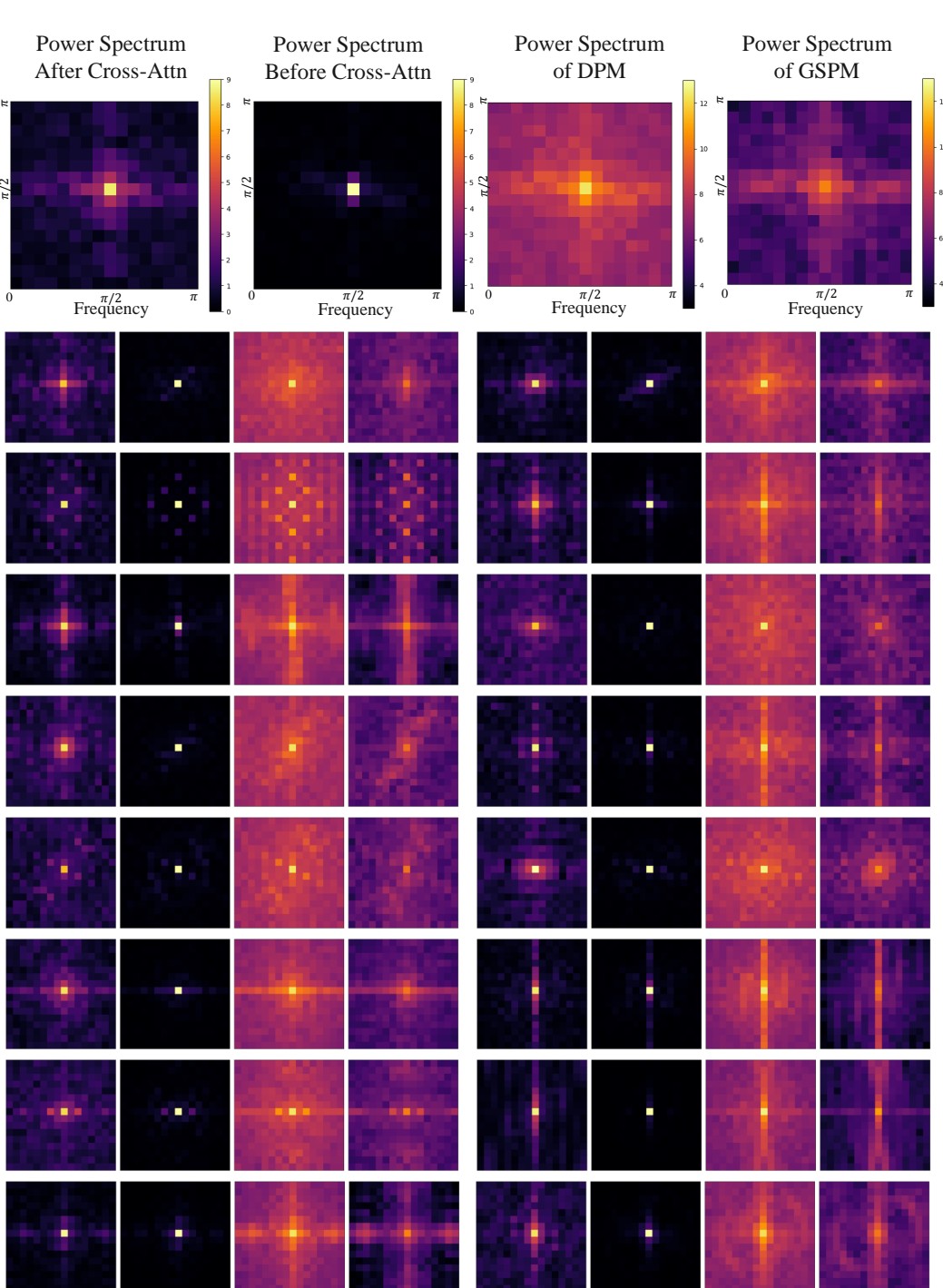

Figure 9: More power spectrum results.

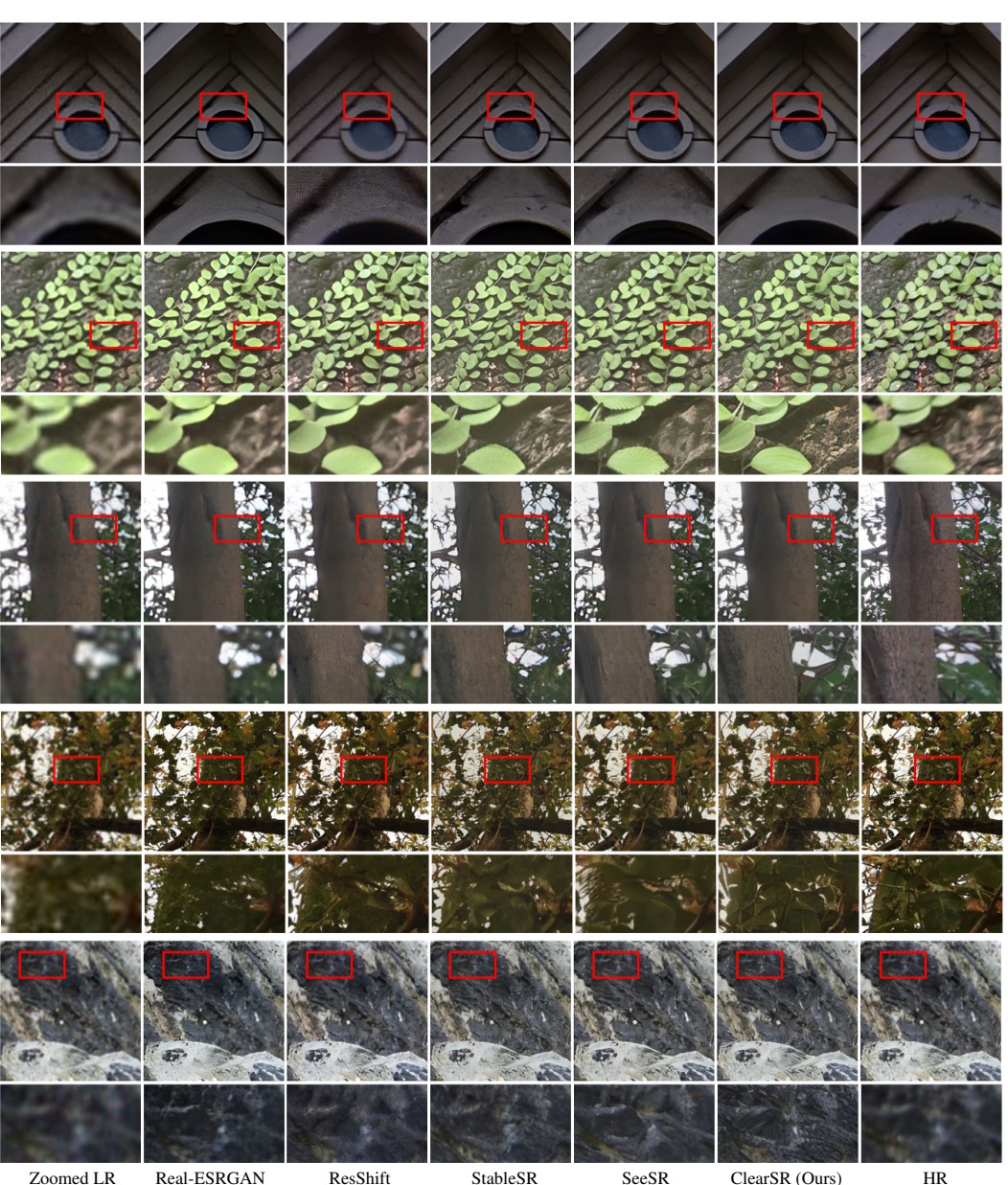

Zoomed LR    Real-ESRGAN    ResShift    StableSR    SeeSR    ClearSR (Ours)    HR

Figure 10: More visual comparisons with recent state-of-the-art Real-ISR methods.

