# OpenReview forum: "ClearSR: Latent Low-Resolution Image Embeddings Help Diffusion-Based Real-World Super Resolution Models See Clearer"
_ICLR.cc/2025/Conference — Submitted to ICLR 2025_

### Official Review · Reviewer_duFd · 2024-10-29

**Soundness:** 3
**Presentation:** 3
**Contribution:** 3
**Rating:** 5
**Confidence:** 5

**Summary:**

This paper introduces ClearSR, a novel method for real-world image super-resolution (Real-ISR) using pretrained T2I diffusion models. ClearSR leverages LR embeddings to constrain ControlNet's control signals,  extracting LR information at detail and structure levels. The authors design DPM and SPM modules, which enhance image details and maintain structural integrity, respectively. Additionally, they propose an LSA strategy during inference to balance fidelity and generative capabilities. Extensive experiments demonstrate that ClearSR outperforms existing methods across multiple benchmarks and metrics.

**Strengths:**

A key challenge in RealSR tasks using powerful T2I models is generating fine details while maintaining fidelity, which presents a trade-off. ClearSR explore this by using a pre-trained VAE encoder as an initial feature extractor for LR images to preserve fidelity as much as possible, and designing DPM and SPM to handle specific control tasks. Additionally, ClearSR observed that the added realistic details largely come from the final inference steps. Therefore, it introduced the LLA mechanism to move away from the LR latent space in the final stages, enhancing generative capability and improving model flexibility.

**Weaknesses:**

1. Lacks more detailed comparisons, such as inference time, parameter count, and computational cost.
2. Missing some key details, like the number of inference steps, and Figure 10 doesn't provide the names of the comparison methods.
3. While the motivation is good, the novelty of the solution seems relatively weak.

**Questions:**

[1] Selection of α and β Parameters:

a. How were the values for α and β in the LSA strategy chosen? Did you perform a systematic parameter search or optimization? Are these parameters required to be tuned for different datasets or image types, and is there a way to automate their selection?

[2] Implementation of LoRA Layers

How does the choice of LoRA rank (set to 16) impact model performance, and was this rank value optimized experimentally?


[3] Something about classfier-free guidance, cfg

During the inference stage, by adjusting the CFG value, RealSR methods based on pre-trained T2I diffusion models can also balance fidelity and perception. The authors did not report the CFG settings during inference, such as the CFG value and negative prompt. Additionally, the proposed LSA control method needs to be compared in detail with the CFG control method to highlight the differences.

If the main concerns are well addressed, I will consider increasing the score.

---

> ### Author Response · Authors · 2024-11-21
>
> Thank you for your valuable feedback. Our response is as follows:
>
> **Q1: Lacks more detailed comparisons, such as inference time, parameter count, and computational cost.**
>
> A1: Thank you for your thoughtful suggestion. We compare the complexity of our ClearSR with that of several SD-based Real-ISR methods (DiffBIR, PASD and SeeSR), including total parameters, trainable parameters, MACs, inference step, inference time and inference speed. All methods are tested on an A40 GPU. Although the additional layers increase the number of parameters and computational cost, we can see that our ClearSR has fewer total parameters, trainable parameters, and MACs compared to SeeSR. For inference speed, since the Diffusers library is optimized for the Classfier-Free Guidance (CFG), we disabled CFG during inference to achieve a fair comparison. Note that DiffBIR originally does not use CFG. In addition, we can also observe that our ClearSR performs well when the inference step is set to 20 (lower MACs and a reduced inference time). This further proves that our model has stronger generative capabilities, allowing it to recover good results even with fewer inference steps. We have added the complexity comparisons to Appendix E in the revision.
>
> | | DiffBIR | PASD | SeeSR | ClearSR | ClearSR-s20 |
> |-----------|-----------|-----------|-----------|-----------|-----------|
> | Total Param (M) | 1717 | 1900 | 2524 | 2511 | 2511 |
> | Trainable Param (M) | 380 | 625 | 750 | 525 | 525 |
> | MACs (G) | 24234 | 29125 | 65857 | 52384 | 21855 |
> | Inference Steps | 50 | 20 | 50 | 50 | 20 |
> | Inference Time (s) | 4.51 | 1.92 | 4.10 | 5.36 | 2.14 |
> | Inference Speed (step/s) | 11.09 | 10.41 | 12.21 | 9.33 | 9.33 |
>
> Quantitative comparison of ClearSR-s20 on DRealSR dataset is as below.
>
> | | PSNR $\uparrow$ | SSIM $\uparrow$ | LPIPS $\downarrow$ | NIQE $\downarrow$ | MUSIQ $\uparrow$ | MANIQA $\uparrow$ | CLIPIQA $\uparrow$ |
> |-----------|-----------|-----------|-----------|-----------|-----------|-----------|-----------|
> |PASD|27.36|0.7073|0.3760|5.5474|64.87|0.6169|0.6808|
> |SeeSR|28.17|0.7691|0.3189|6.3967|64.93|0.6042|0.6804|
> |ClearSR|28.22|0.7538|0.3473|6.0867|66.27|0.6246|0.6976|
> |ClearSR-s20|28.53|0.7689|0.3543|7.4823|65.88|0.6088|0.7176|
>
> **Q2: Missing some key details, like the number of inference steps, and Figure 10 doesn't provide the names of the comparison methods.**
>
> A2: Thank you for your reminder. The number of inference steps is 50. The comparison methods in Figure 10 are the same as those in Figure 6, listed from left to right as follows: Zoomed LR, Real-ESRGAN, ResShift, StableSR, SeeSR, ClearSR (Ours), HR. We have made this clear in the revision.
>
> **Q3: While the motivation is good, the novelty of the solution seems relatively weak.**
>
> A3: Thank you for recognizing the motivation of our paper. Our solution primarily focuses on making the model’s generation more consistent with the LR information. To achieve this, we use additional cross-attention layers to further constrain the control signal and design two modules to preserve detail and structural information.
> In terms of the solution, firstly, our solution is novel. To our knowledge, we are the first to highlight the importance of LR latent embedding. We efficiently utilize LR information by providing additional constraints in the latent space to obtain better control signals. Previous methods, such as PASD, did not optimize the control signal itself. SeeSR uses the semantic signals to improve the model's generative capability, which might lead to outputs that are inconsistent with the LR information. Moreover, our approach is simple and effective. In Figure 2, we show that our method can better extract LR information. In Figure 4, we can see that the output from DPM contains more high-frequency information which is helpful for reconstructing details while SPM mainly contains low-frequency information which preserves structural information.
> However, there are still some limitations in our approach. For instance, when constraining the control signal, we hope to design more efficient solutions. Additionally, in the decoupling of high-frequency and low-frequency information, it may be necessary to supply further processed LR information to the DPM and SPM. This will be part of our future work.

---

> ### Author Response · Authors · 2024-11-21
>
> **Q4: Selection of α and β Parameters: How were the values for α and β in the LSA strategy chosen? Did you perform a systematic parameter search or optimization? Are these parameters required to be tuned for different datasets or image types, and is there a way to automate their selection?**
>
> A4: Thank you for your valuable question.
> 1. "How were the values for α and β in the LSA strategy chosen? Did you perform a systematic parameter search or optimization?"
> Our LSA method is designed to balance fidelity and generation. However, the "optimal balance" between fidelity and generation is relatively subjective. In the paper, we tested multiple sets of α and β values and ultimately selected the ones that we believed were relatively well balanced in fidelity and generation.
> 2. "Are these parameters required to be tuned for different datasets or image types?"
> Since our model performs well under various degradation conditions, these parameters do not need to be tuned for different datasets or image types.
> As mentioned in the original paper, we used LoRA layers to fine-tune the VAE, enabling our model to adapt to severe degradation conditions. Please note that during inference, we also use the fine-tuned VAE to provide LR guidance. We will conduct an experiment to validate our model's ability to adapt to degradation.
> We conducted the experiment on the DRealSR dataset, where we added extra degradation to the LR images to simulate severe degradation conditions. Using the HR image as the reference, we calculated the PSNR, SSIM, and LPIPS before and after adding degradation:
>
> || PSNR $\uparrow$ | SSIM $\uparrow$| LPIPS$ \downarrow$ |
> |-----------|-----------|-----------|-----------|
> |Before adding degradations|30.57|0.8301|0.4608|
> |After adding degradations|29.03|0.7961|0.5698|
>
> Subsequently, we input the degraded images into SeeSR and ClearSR. The table below shows the metrics before and after adding degradation:
>
> || PSNR $\uparrow$ | SSIM $\uparrow$ | NIQE $\downarrow$ | MANIQA $\uparrow$ |
> |-----------|-----------|-----------|-----------|-----------|
> |SeeSR before|28.17|0.7691|6.3967|0.6042|
> |SeeSR after|27.47|0.7525|6.6463|0.6028|
> |ClearSR before|28.22|0.7538|6.0867|0.6246|
> |ClearSR after|27.74|0.7391|6.3062|0.6221|
>
> We further calculated the changes in these metrics before and after adding degradation:
>
> | | Δ PSNR $\uparrow$ | Δ SSIM $\uparrow$ | Δ NIQE $\downarrow$ | Δ MANIQA $\uparrow$ |
> |-----------|-----------|-----------|-----------|-----------|
> |SeeSR|-0.70|-0.0166|+0.2496|-0.0014|
> |ClearSR|-0.48|-0.0147|+0.2195|-0.0025|
>
> We can see that the generative metric changes for SeeSR and ClearSR are relatively small. However, ClearSR shows a smaller decrease in PSNR and SSIM, indicating that it adapts better to severe degradation conditions.
> 3. "Is there a way to automate their selection?"
> Automatically selecting hyperparameters is a complex task that typically requires extensive engineering investigation to determine the final solution. Therefore, we did not report these results in the paper. However, we proposed a simple and effective strategy to validate the feasibility of automatic hyperparameter selection:
> For instance, by first calculating the MANIQA score of the LR image and then choosing different values of α and β based on the MANIQA score. Our strategy is demonstrated in the table below:
>
> |Group|MANIQA=m|α|β|
> |-----------|-----------|-----------|-----------|
> |1|m<0.35|0.000|0.015|
> |2|0.35≤m＜0.45|0.005|0.010|
> |3|0.45≤m<0.55|0.010|0.005|
> |4|0.55≤m|0.015|0.000|
>
> In the table below, we present the metrics using this strategy:
>
> || PSNR $\uparrow$ | SSIM $\uparrow$ | NIQE $\downarrow$ | MANIQA $\uparrow$ |
> |-----------|-----------|-----------|-----------|-----------|
> |ClearSR base|28.22|0.7538|6.0867|0.6246|
> |ClearSR auto|28.26|0.7552|6.2290|0.6221|
>
> Although our LSA method performs similarly under various degradation conditions, meaning this strategy may not improve overall metrics, it does have specific effects. We present the metrics for each group of LR images below. We can see that for lower-quality LR images (Group 1, Group 2), our model tends to generate more details (as shown by improvements in NIQE and MANIQA). For higher-quality LR images (Group 3, Group 4), the model outputs results more consistent with the LR image (as shown by improvements in PSNR and SSIM). Overall, using this strategy enhances the visual quality of the model's output.
>
> |Group|MANIQA=m| PSNR $\uparrow$ | SSIM $\uparrow$ | NIQE $\downarrow$ | MANIQA $\uparrow$ |
> |-----------|-----------|-----------|-----------|-----------|-----------|
> |1 base|m<0.35|25.82|0.7205|6.9769|0.6444|
> |1 auto|m<0.35|25.31|0.6919|6.5023|0.6681|
> |2 base|0.35≤m＜0.45|29.35|0.7989|6.8092|0.6034|
> |2 auto|0.35≤m＜0.45|29.15|0.7925|6.6800|0.6128|
> |3 base|0.45≤m<0.55|28.18|0.7431|5.6731|0.6257|
> |3 auto|0.45≤m<0.55|28.29|07469|5.8528|0.6222|
> |4 base|0.55≤m|26.69|0.7136|6.0585|0.6583|
> |4 auto|0.55≤m|27.03|0.7279|6.7394|0.6443|

---

> ### Author Response · Authors · 2024-11-21
>
> **Q5: Implementation of LoRA Layers: How does the choice of LoRA rank (set to 16) impact model performance, and was this rank value optimized experimentally?**
>
> A5: Thank you for your question. We select the appropriate LoRA rank experimentally.
> The table below shows the ablation results for the VAE LoRA rank. As seen, reducing LoRA rank improves fidelity but has a negative impact on generation metrics. On the contrary, increasing LoRA rank has a negative impact on fidelity but improves generation metrics. To balance fidelity and generation, we finally set the VAE LoRA rank to 16.
>
> | VAE LoRA rank| PSNR $\uparrow$ | SSIM $\uparrow$ | MUSIQ $\uparrow$ | MANIQA $\uparrow$ |
> |-----------|-----------|-----------|-----------|-----------|
> |8|27.70|0.7512|66.64|0.6221|
> |16|27.62|0.7483|66.80|0.6222|
> |32|27.46|0.7346|67.06|0.6311|
>
> The table below shows the ablation results for the UNet LoRA rank. We can see that both smaller LoRA rank and larger LoRA rank improve fidelity but have a negative impact on generation metrics. To balance fidelity and generation, we finally set the UNet LoRA rank to 16. We have added this ablation study to Appendix F in the revision.
>
> | UNet LoRA rank| PSNR $\uparrow$ | SSIM $\uparrow$ | MUSIQ $\uparrow$ | MANIQA $\uparrow$ |
> |-----------|-----------|-----------|-----------|-----------|
> |8|27.70|0.7508|66.28|0.6165|
> |16|27.62|0.7483|66.80|0.6222|
> |32|27.83|0.7533|66.35|0.6166|
>
> **Q6: Something about classfier-free guidance, cfg: During the inference stage, by adjusting the CFG value, RealSR methods based on pre-trained T2I diffusion models can also balance fidelity and perception. The authors did not report the CFG settings during inference, such as the CFG value and negative prompt. Additionally, the proposed LSA control method needs to be compared in detail with the CFG control method to highlight the differences.**
>
> A6: Thank you for your thoughtful suggestion. Similar to PASD and SeeSR, we also utilized Classifier-Free Guidance (CFG). The cfg scale is set to 7, the positive prompt is "clean, high-resolution, 8k, detailed, realistic", and the negative prompt is "dotted, noise, blur, lowres, smooth".
> Regarding the differences between LSA and CFG, our analysis is as follows:
> 1. In terms of approach: LSA directly adjusts the prediction through LR latent embeddings, while CFG controls generation by blending predictions from two different prompts.
> 2. In terms of computational cost: LSA has almost no extra computational cost, while CFG requires generating two predictions of two prompts, which doubles the computational cost.
> 3. In terms of adjustment range: LSA can enhance the fidelity by increasing α and can also enhance the generative capability of the model by increasing β, serving as a bidirectional adjustment strategy. With CFG, a given prompt set can only adjust in one direction. Additionally, setting the cfg scale too high often results in unnatural outputs, limiting its adjustment range.
> 4. In terms of the output quality: Using ClearSR settings as a baseline, we adjusted the model's output through both LSA and CFG, aiming for a PSNR of 29 for high fidelity and a PSNR of 27.5 for high generative capability. In the first comparison, results using LSA have advantages on generation metrics. In the second comparison, although the generative metrics were similar, an excessively high cfg scale led to unnatural outputs, reflected by a significant increase in LPIPS.
>
> | | PSNR $\uparrow$ | SSIM $\uparrow$ | LPIPS $\downarrow$ | MANIQA $\uparrow$ |
> |-----------|-----------|-----------|-----------|-----------|
> |cfg scale = 2|28.95|0.7721|0.3278|0.5767|
> |α=0.03, β=0.01|29.00|0.7781|0.3281|0.5878|
>
> | | PSNR $\uparrow$ | SSIM $\uparrow$ | LPIPS $\downarrow$ | MANIQA $\uparrow$ |
> |-----------|-----------|-----------|-----------|-----------|
> |cfg scale = 12|27.51|0.7336|0.3719|0.6507|
> |α=0.01, β=0.04|27.45|0.7271|0.3415|0.6511|
>
> Overall, LSA outperforms CFG in terms of computational cost, adjustment range, and the output quality.

---

> > ### Comment · Reviewer_duFd · 2024-11-22
> >
> > Thank you for the author’s reply. Some of my concerns have been addressed, but there are still issues that need clarification.
> >
> > **Regarding my raised weakness (3): While the motivation is good, the novelty of the solution seems relatively weak.**
> >
> >
> > The author claims their approach is novel, but I remain skeptical for the following reasons:
> >
> > 1. The control mechanisms involving cross-attention operations and add operations have already been proposed in PASD and ControlNet.
> >
> > 2. The design principles of the DPM and SPM modules do not appear to be particularly distinctive. Why can one help restore details while the other restores structures? Although the authors provide some explanations from the perspective of power spectrum analysis, this might be a hand-picked result rather than a general case. It is necessary to differentiate the functional roles of these two modules based on design principles rather than outcome-based reasoning.
> >
> > 3. The LSA strategy is derived from PASD and SUPIR. The Early-step LR Adjustment (ELA) is similar to the Adjustable Noise Schedule (ANS) in PASD, as both suppress overgeneration by adjusting the LR mixing ratio in the early diffusion steps. The Later-step LR Adjustment (LLA) resembles the restoration-guided sampling strategy in SUPIR, as both enhance detail generation by reducing the LR ratio during later diffusion steps.

---

> > > ### Author Response · Authors · 2024-11-23
> > >
> > > Thank you for your reply. We are glad that we were able to address some of your concerns. We will provide a detailed discussion of weakness (3).
> > >
> > > **Q7. The control mechanisms involving cross-attention operations and add operations have already been proposed in PASD and ControlNet.**
> > >
> > > A7: Thank you for your reminder.
> > >
> > > Regarding cross-attention, our design principle differs fundamentally from PASD. PASD introduces the PACA, which allows the control signal, before passing through the zero convolution layer, to directly interact with the features in the UNet. The goal of this design is to better integrate the control signal into the UNet. However, it does not improve the control signal itself. In contrast, as shown in Figure 2 of the ClearSR paper, we observed that the control signal provided by the original ControlNet has a bias relative to the LR latent embedding. Strengthening the utilization of such biased control signals still cannot provide accurate guidance to the UNet, which limits the model's potential.
> > >
> > > In contrast, our DPM is built on ControlNet, with the addition of cross-attention layers to constrain the control signal itself. This reflects our design principle that improving the quality of the control signal could provide accurate guidance to the UNet, which allows the model to reach a higher potential, which represents a new paradigm in this field.
> > >
> > > Moreover, SeeSR also uses cross-attention layers to integrate semantic signal. However, the semantic signal is also biased. As shown in Figure 1 of the ClearSR paper, this semantic signal might lead to inconsistent generation with the LR image. This means that while SeeSR enriches the information contained in the control signal, it might have a negative impact due to the bias in the semantic signal.
> > >
> > > Regarding the add operations, in our DPM, we did not follow ControlNet's add operation to perform the noise addition process. Instead, we adopted DiffBIR's concatenation approach. The noise is concatenated with the LR condition, which means the impact of randomness brought by the noise is reduced. When combining the control signals provided by DPM and SPM, we used the add operation. However, this is an experimental result. We tried various strategies to combine, including designing additional MLP layers and using scales related to the timestep. The results of this ablation study are as below.
> > >
> > > | | PSNR $\uparrow$ | SSIM $\uparrow$ | LPIPS $\downarrow$ | NIQE $\downarrow$ | MUSIQ $\uparrow$ | MANIQA $\uparrow$ | CLIPIQA $\uparrow$ |
> > > |-----------|-----------|-----------|-----------|-----------|-----------|-----------|-----------|
> > > |ClearSR-scale related to timestep|28.24|0.7557|0.3530|6.1802|66.84|0.6285|0.6907|
> > > |ClearSR-additional MLP layers|28.31|0.7495|0.3579|6.0696|66.43|0.6203|0.6952|
> > > |ClearSR|28.22|0.7538|0.3473|6.0867|66.27|0.6246|0.6976|
> > >
> > > It can be seen that the impact of these different strategies on the model's performance is quite slight. As a result, we finally choose the simple but effective add operation.

---

> > > ### Author Response · Authors · 2024-11-23
> > >
> > > **Q8. The design principles of the DPM and SPM modules do not appear to be particularly distinctive. Why can one help restore details while the other restores structures? Although the authors provide some explanations from the perspective of power spectrum analysis, this might be a hand-picked result rather than a general case. It is necessary to differentiate the functional roles of these two modules based on design principles rather than outcome-based reasoning.**
> > >
> > > A8: Apologies for the confusion caused by our description. First, we will clarify the difference between the DPM and SPM.
> > >
> > > The purpose of the DPM is to preserve details. To achieve this, our DPM is based on ControlNet and introduces additional window-based cross-attention layers to constraint the control signal in the latent space. These cross-attention layers are placed after text cross-attention layers. The DPM is initialized using the parameters of the UNet, with a total parameter count of 436M.
> > >
> > > The purpose of the SPM is to preserve structural information. To extract global structural information, a deep network is not necessary. Therefore, the SPM only retains 4 Resblocks and 3 Downsamples from the ControlNet to provide control signals that match the shape of the control signals provided by the DPM. The DPM is trained from scratch, with a total parameter count of 84M.
> > >
> > > We then analyze why the DPM could help restore details while the SPM restores structures. From a structural perspective, the DPM is deeper and contains number of attention layers. Compared to convolutional layers, these attention layers are more effective at preserving detailed information. In contrast, the SPM is shallower and does not contain any attention layers. This structure makes it difficult for SPM to learn the complex, deep features in the LR image and instead tends to preserve simple global structural features.
> > >
> > > Furthermore, from the perspective of parameter initialization, the DPM uses UNet's parameter initialization, which means it benefits from the generative priors of UNet to help restore details. The SPM does not use UNet's parameter initialization, and therefore cannot use generative priors to help generate details.
> > >
> > > We apologize for any confusion caused by our power spectrum analysis. This is not a hand-picked result, but a general outcome. The 18 images shown in Figure 4 and Figure 9 of the ClearSR paper are all from the DRealSR test set (which contains 93 images). Additionally, we average the power spectrum (16*16) of each feature and present the results in the table below. The average values for the entire test set are: DPM=7.4823, SPM=5.3395. This value represents the logarithm of the power spectrum, with higher values indicating more high-frequency components. As can be seen, our power spectrum analysis represents a general case.

---

> > > ### Author Response · Authors · 2024-11-23
> > >
> > > |Image|DPM|SPM|Image|DPM|SPM|Image|DPM|SPM|Image|DPM|SPM|
> > > |-----------|-----------|-----------|-----------|-----------|-----------|-----------|-----------|-----------|-----------|-----------|-----------|
> > > |Canon_10| 7.4837| 5.8041|Canon_14| 7.1098| 4.7168|Canon_40| 7.3121| 5.2213|Canon_42| 6.8476| 4.2477|
> > > |Canon_56| 7.0225| 4.7880|DSC_0988| 6.9629| 4.6750|DSC_1045| 7.3681| 5.5637|DSC_1057| 7.4462| 5.4421|
> > > |DSC_1137| 7.7509| 5.8143|DSC_1233| 7.7329| 6.0554|DSC_1241| 7.3576| 5.4401|DSC_1245| 7.4513| 5.2182|
> > > |DSC_1265| 7.4860| 5.3653|DSC_1286| 7.4396| 5.0683|DSC_1326| 7.7430| 5.7232|DSC_1404| 7.1466| 4.9589|
> > > |DSC_1412| 7.0540| 4.4695|DSC_1425| 7.8256| 6.0219|DSC_1454| 7.7057| 5.5883|DSC_1462| 7.8962| 5.9595|
> > > |DSC_1474| 7.7036| 5.8552|DSC_1575| 7.7683| 5.7031|DSC_1583| 7.8893| 6.0474|DSC_1599| 7.7467| 5.6074|
> > > |DSC_1603| 7.8664| 5.9849|IMG_107| 7.0049| 3.9978|IMG_113| 7.0470| 4.6107|IMG_118| 7.2686| 5.2510|
> > > |IMG_125| 7.4606| 5.2799|IMG_130| 7.8483| 5.9704|IMG_140| 7.5892| 5.5650|IMG_143| 7.9063| 5.8560|
> > > |IMG_150| 7.1805| 4.3383|IMG_181| 7.9043| 5.9429|IMG_190| 7.6954| 5.4518|IMG_203| 7.5813| 5.9380|
> > > |IMG_210| 7.3911| 5.5596|P1140090| 7.4658| 5.1093|P1140122| 7.4762| 4.8260|P1140126| 7.6926| 5.2176|
> > > |P1140134| 7.2462| 4.8982|P1140138| 7.5991| 5.0509|P1140177| 7.5900| 5.0371|P1140189| 7.2313| 4.6990|
> > > |P1140279| 7.7133| 5.4946|P1140388| 7.2887| 4.8450|P1140401| 7.1972| 4.8202|P1140417| 7.5324| 5.1514|
> > > |P1140434| 7.6094| 5.4017|P1160566| 6.2740| 3.4998|P1160646| 7.2306| 5.0274|P1160772| 7.7088| 5.8862|
> > > |P1160776| 7.2626| 5.8510|P1171010| 7.7851| 5.6909|P1171031| 7.6059| 5.3664|P1171051| 7.5015| 5.3215|
> > > |panasonic_103| 7.2776| 5.1857|panasonic_123| 7.8734| 5.8466|panasonic_128| 7.2675| 5.4307|panasonic_132| 7.3756| 5.2387|
> > > |panasonic_145| 7.5162| 5.2170|panasonic_158| 7.4802| 5.6096|panasonic_16| 7.0668| 5.0267|panasonic_182| 7.4215| 5.5394|
> > > |panasonic_187| 7.2840| 5.1467|panasonic_196| 7.6008| 5.2809|panasonic_197| 7.8238| 5.7726|panasonic_202| 7.7592| 5.7576|
> > > |panasonic_206| 7.7062| 5.5134|panasonic_233| 7.7203| 5.8975|panasonic_43| 7.7803| 5.7019|panasonic_50| 7.4098| 5.2150|
> > > |panasonic_57| 7.5528| 5.2498|panasonic_60| 7.4726| 5.5752|panasonic_62| 6.9740| 3.9904|panasonic_85| 7.3339| 4.5893|
> > > |sony_1| 7.5265| 5.6137|sony_100| 7.7961| 5.9548|sony_107| 7.5842| 5.5240|sony_11| 7.5412| 5.5975|
> > > |sony_116| 7.8659| 5.8117|sony_120| 7.8855| 6.0696|sony_129| 7.7181| 5.4436|sony_158| 6.9894| 4.8268|
> > > |sony_160| 6.9831| 5.0128|sony_169| 7.9072| 6.1190|sony_183| 7.5594| 5.7758|sony_189| 6.7753| 4.0597|
> > > |sony_49| 7.4994| 5.2796|sony_53| 7.7342| 5.7925|sony_54| 7.6660| 5.5743|sony_80| 7.6056| 5.4975|
> > > |sony_82| 7.5186| 5.5391|

---

> > > ### Author Response · Authors · 2024-11-23
> > >
> > > **Q9. The LSA strategy is derived from PASD and SUPIR. The Early-step LR Adjustment (ELA) is similar to the Adjustable Noise Schedule (ANS) in PASD, as both suppress overgeneration by adjusting the LR mixing ratio in the early diffusion steps. The Later-step LR Adjustment (LLA) resembles the restoration-guided sampling strategy in SUPIR, as both enhance detail generation by reducing the LR ratio during later diffusion steps.**
> > >
> > > A9: Apologies for the confusion caused by our description of the LSA strategy. We will provide a detailed explanation of the differences compared to other methods.
> > >
> > > Our LSA strategy is derived from observations and analysis of the inference stage. Based on CCSR[1], we divide the entire inference stage into three parts and apply different adjustment strategies for the early and later steps. The difference is that PASD and SUPIR did not conduct in-depth research on the inference stage. PASD only studied adjustments for the early stage, while SUPIR uses the same LR latent addition strategy for the entire inference stage.
> > >
> > > Our ELA differs fundamentally from ANS in both approach and design purpose. ANS attempts to eliminate the inconsistency in residual signals between training and testing by adding LR latent to the initial Gaussian noise. In contrast, ELA attempts to provide direct adjustment in the latent space by using the LR latent to adjust the predicted $\mathbf{x_0}$ at each step.
> > >
> > > Our LLA also differs from the restoration-guided sampling strategy. SUPIR still aims for each prediction in the later steps to **be closer to** the LR latent. However, based on our division of the inference stage, in the later steps, the model mainly focuses on detail enhancement. In contrast to SUPIR, we leverage the inherent properties of the LR image, which mainly contains structural information and has fewer details compared to HR images, to adjust the predicted $\mathbf{x_0}$ in the opposite direction, which means let $\mathbf{x_0}$ **move away from** the LR latent. Essentially, this provides guidance to the model, allowing it to generate more details in the appropriate regions. This guidance allows our LLA to improve the model's generative capabilities.
> > >
> > > Additionally, SUPIR reduces the LR ratio in the later diffusion steps, but this only weakens the effect of the restoration-guided sampling strategy, and does not improve generative capabilities compared to the original model. As shown in the table below, when set α and β to 0.00 and 0.01, our ClearSR achieves higher generative metrics compared to the original model (α=0.00, β=0.00), which is something that SUPIR’s strategy cannot achieve.
> > >
> > > |α|β| PSNR $\uparrow$ | SSIM $\uparrow$ | NIQE$\downarrow$ | MANIQA $\uparrow$ |
> > > |-----------|-----------|-----------|-----------|-----------|-----------|
> > > |0.00|0.00| 28.11| 0.7419|6.5289|0.6226|
> > > |0.01|0.00| 28.44| 0.7609|6.4765|0.6172|
> > > |0.00|0.01| 27.85| 0.7412|5.9875|0.6360|
> > > |0.01|0.01| 28.22| 0.7538|6.0867|0.6246|
> > >
> > > [1]Lingchen Sun, Rongyuan Wu, Zhengqiang Zhang, Hongwei Yong, and Lei Zhang. Improving the stability of diffusion models for content consistent super-resolution. arXiv preprint arXiv:2401.00877, 2023.

---

> > > > ### Comment · Reviewer_duFd · 2024-11-24
> > > >
> > > > Response to A7,A8,A9
> > > >
> > > > **Response to A7:**
> > > >
> > > > Thank you for the author's explanation.
> > > >
> > > > 1. My primary concern pertains to the technical advancements in the control method. The author employs a cross-attention mechanism to regulate the DPM module, which closely resembles the PACA method proposed in PASD. For the UNet Decoder, the author adopts the same "add" operation as ControlNet. Additionally, ClearSR utilizes the LR latent encoded by the VAE encoder as the input for the control component (similar to StableSR). **Finally, ClearSR applies the LR latent through a cross-attention mechanism to control the DPM module. This distinction from prior works appears to represent only a minor technical improvement.**
> > > >
> > > > 2. A point of confusion arises from the fact that the DPM module, which is purportedly designed to enhance details, while integrates more LR latent information into its process. From a general perspective, increasing constraints from LR typically results in degraded generative performance [1,2]. The author is encouraged to clarify the rationale behind this design choice.
> > > >
> > > > [1] Scaling up to excellence: Practicing model scaling for photo-realistic image restoration in the wild
> > > >
> > > > [2] Seesr: Towards semantics-aware real-world image super-resolution
> > > >
> > > > **Response to A8:**
> > > >
> > > > Thank you for the reviewer’s clarification. Please refer to **Response to A7 (2)** for my questions regarding the DPM module.
> > > >
> > > > **Response to A9:**
> > > >
> > > > Thank you for the reviewer’s clarification. I created a table to present the similarities and differences between ClearSR and SUPIR in terms of latent control strategies. ClearSR adopts a similar strategy to SUPIR, suppressing over-generation by remaining close to the LR latent. In the later stages, however, ClearSR introduces a novel approach by moving away from the LR latent to further enhance generative capability. Nonetheless, the authors should clarify the relationship between the ELA strategy and the restoration-guided sampling strategy proposed by SUPIR in the paper.
> > > >
> > > > |                       | **SUPIR**                  | **ClearSR**                 |
> > > > |:----------------------:|:--------------------------:|:---------------------------:|
> > > > | **early diffusion stage** | (1 - $\alpha$)x + $\alpha$ LR    | (1 - $\alpha$)x + $\alpha$ LR     |
> > > > | **later diffusion stage** | (1 - $\alpha$)x + $\alpha$ LR    | (1 + $\beta$)x - $\beta$ LR       |

---

> > > > > ### Author Response · Authors · 2024-11-25
> > > > >
> > > > > Thank you for your reply. Regarding your point, here is our further discussion:
> > > > >
> > > > > **Q10: My primary concern pertains to the technical advancements in the control method. The author employs a cross-attention mechanism to regulate the DPM module, which closely resembles the PACA method proposed in PASD. For the UNet Decoder, the author adopts the same "add" operation as ControlNet. Additionally, ClearSR utilizes the LR latent encoded by the VAE encoder as the input for the control component (similar to StableSR). Finally, ClearSR applies the LR latent through a cross-attention mechanism to control the DPM module. This distinction from prior works appears to represent only a minor technical improvement.**
> > > > >
> > > > > A10: Thank you for your valuable question.
> > > > >
> > > > > Firstly, **our ClearSR primarily focuses on the utilization of LR latent embeddings**. In previous works, the importance of LR latent embeddings has often been overlooked. To the best of our knowledge, we are the first to propose this perspective and provide a solution. **Our main goal is to find methods that can more effectively leverage LR latent embeddings to provide a better control signal** which is mentioned in the response to Q7. Building on this, we designed DPM and SPM to use LR latent embeddings to enhance the control signal, thereby allowing the model to reach a higher potential, which represents a new paradigm in this field.
> > > > >
> > > > > Next, We will compare our method with other methods **from the perspective of technical advancements mainly**.
> > > > >
> > > > > **Regarding the cross-attention mechanism**: As mentioned in response to Q7, our window-based cross-attention layers are placed in the control branch, while the PACA layers are placed in the UNet. This reflects our different design principles: we aim to improve the quality of the control signal itself, while PASD seeks to make better use of the control signal. Although the cross-attention mechanism is a common and effective technique to combine additional information, we have still optimized it. We use window partition to better aggregate local information, and we demonstrate its effectiveness. The effectiveness of window partitioning is shown in Table 2 of the paper, and the ablation study regarding the window size is provided below. **To the best of our knowledge, no previous SD-based RealISR method has combined information in this way.**
> > > > >
> > > > > | window size| PSNR $\uparrow$ | SSIM $\uparrow$ | NIQE $\downarrow$ | MANIQA $\uparrow$ |
> > > > > |-----------|-----------|-----------|-----------|-----------|
> > > > > |32|27.29|0.7294|6.5364|0.6333|
> > > > > |16(ClearSR)|27.62|0.7483|6.6334|0.6222|
> > > > > |8|27.97|0.7619|6.9919|0.6090|
> > > > >
> > > > > **Regarding the "add" operation**: Since our control branch is based on ControlNet, this is the standard way of integrating the control signal into the UNet. As mentioned in our response to Q7, our design principle focuses on obtaining a better control signal. **How to integrate the control signal is not the main focus of our work. Furthermore, our ClearSR is not in conflict with these strategies (such as PASD’s PACA and SUPIR’s ZeroSFT).** We may incorporate these modules in future versions of ClearSR to achieve better performance.
> > > > >
> > > > > **Regarding the VAE encoder**: Since we need to correctly map the LR image to the latent space, it is generally necessary to use an appropriate encoder to perform this process. Using the pre-trained VAE from Stable Diffusion is a common way (such as in DiffBIR and SUPIR). However, because the pre-trained VAE is usually not well-suited for the degradation of LR images, some modifications are required, such as adding denoising modules (as in DiffBIR), fine-tuning the pre-trained encoder (as in SUPIR), or manually designing a new encoder (as in SeeSR). Regardless, the goal of these methods is to ensure that the LR image is correctly embedded into the latent space.
> > > > >
> > > > > In the ClearSR paper, we propose an efficient method for fine-tuning the VAE. We achieve this by simply adding LoRA layers during training. This approach does not require the design of additional denoising modules (as in DiffBIR), does not require an extra training phase (as in SUPIR), does not require the manual design of the encoder structure (as in SeeSR), and incurs almost no additional training parameters, which we consider a technical advancement.
> > > > >
> > > > > **Regarding other technical advancement**: As mentioned above, our window partition strategy in DPM is an effective method for integrating LR latent information, and the efficient fine-tuning strategy for the VAE that we propose is an effective way to correctly map the LR image to the latent space. These reflect the technical advancements in our control method. Additionally, as mentioned in the response to Q8, our SPM is an efficient module for extracting structural information from the LR image to optimize the control signal, and it is also a technical advancement.

---

> > > > > ### Author Response · Authors · 2024-11-25
> > > > >
> > > > > **In summary**, as the common use of the "add" operation, cross-attention mechanism, and VAE encoder in previous methods, we believe that using these methods is reasonable as we have different design principles. Furthermore, our technical advancements are reflected in the window partition, VAE fine-tuning, and the design of SPM.
> > > > >
> > > > > We apologize for any confusion our descriptions may have caused. If you have any further questions, please feel free to submit a new response to us anytime.
> > > > >
> > > > > **Q11: A point of confusion arises from the fact that the DPM module, which is purportedly designed to enhance details, while integrates more LR latent information into its process. From a general perspective, increasing constraints from LR typically results in degraded generative performance [1,2]. The author is encouraged to clarify the rationale behind this design choice.**
> > > > >
> > > > > A11: Thank you for your valuable question. Integrating more LR latent information into the control signal and integrating more LR latent information into the predicted $\mathbf{x_0}$ are different.
> > > > >
> > > > > Using the LR latent to adjust the predicted $\mathbf{x_0}$ does result in degraded generative performance. However, as mentioned in Section 3.1 of the SeeSR paper, the purpose of introducing semantic information is to better leverage the generative priors, which demonstrates that the control signal's role is more about guiding the generative priors.
> > > > >
> > > > > In this term, our control signal combines more LR latent information, allowing it to provide more accurate guidance to the UNet. This helps enhance the generative ability of the model.
> > > > >
> > > > > **Q12: The authors should clarify the relationship between the ELA strategy and the restoration-guided sampling strategy proposed by SUPIR in the paper.**
> > > > >
> > > > > A12: Thank you for your thoughtful suggestion. We will compare our ELA with the restoration-guided sampling (RGS) strategy proposed by SUPIR in detail.
> > > > >
> > > > > Our ELA differs from SUPIR both in purpose and approach.
> > > > >
> > > > > **Regarding the purpose**: The goal of SUPIR is to limit the generation to ensure that the image recovery is faithful to the LQ image.
> > > > >
> > > > > Our method is based on the observations from CCSR. CCSR focuses on improvements during the training phase, while we focus on improving the inference stage and divide it into three parts: Structure Refinement, Content Generation, and Detail Enhancement. In this term, the goal of our ELA is to help the model perform better in the **Structure Refinement** stage, while our LLA aims to assist the model in the **Detail Enhancement** stage. Therefore, **the purpose of our LSA is to provide adaptive guidance tailored to the different stages of the inference process**.
> > > > >
> > > > > **Regarding the approach**: Although our ELA uses a similar latent space adjustment strategy as SUPIR, there are key differences. SUPIR applies the same adjustment strategy throughout the entire inference stage, with α changing at each inference step. In contrast, our ELA only adjusts $\mathbf{x_0}$ in the early steps and uses a fixed α (based on our analysis of NIQE, the early steps refer to the first 12 inference steps). According to our division of the inference stage, during the **Content Generation** phase, we do not apply any adjustments.
> > > > >
> > > > > We conducted an experiment to verify the necessity of our division of the inference stage. When the predicted $$\mathbf{x_0$$ is adjusted during the **Content Generation** phase, the model's generative capability tends to be lost more.
> > > > >
> > > > > |steps for ELA|α|steps for LLA|β| PSNR $\uparrow$ | SSIM $\uparrow$ | NIQE$\downarrow$ | MANIQA $\uparrow$ |
> > > > > |-----------|-----------|-----------|-----------|-----------|-----------|-----------|-----------|
> > > > > |1-30| 0.01 | 46-50| 0.01 | 29.31 | 0.7934 | 8.7730 | 0.5424 |
> > > > > |1-12| 0.045| 46-50| 0.01| 29.38 | 0.7913 | 7.8038 | 0.5508 |
> > > > > |1-12| 0.01 | 30-50 | 0.01 | 27.50| 0.7239 | 6.4406 | 0.6383 |
> > > > > |1-12 | 0.01| 46-50 | 0.035|27.59| 0.7321 | 6.1003 | 0.6479 |
> > > > >
> > > > > As shown in the first and second rows, the first row applies ELA over a larger range, while the second row adjusts the α value in LSA to achieve a similar PSNR result. It can be observed that the first row has lower generative metrics compared to the second row. In the third and fourth rows, the third row applies LLA over a larger range, while the fourth row adjusts the β value in LSA to achieve a similar PSNR result. Similarly, the third row shows lower generative metrics compared to the fourth row.
> > > > >
> > > > > Additionally, in RGS, a smaller α is used in the later stage to mitigate the issue we mentioned above. However, this approach only alleviates the problem and does not explicitly use different strategies for different stages based on an analysis of the inference process.
> > > > >
> > > > > Based on the above results, it can be concluded that our ELA approach for integrating LR latent is more reasonable than SUPIR’s method.
> > > > >
> > > > > If our explanation still confuses you, please feel free to submit a new response to us anytime.

---

> > > > > ### Author Response · Authors · 2024-12-01
> > > > >
> > > > > Dear Reviewer duFd,
> > > > >
> > > > > Thank you once again for your valuable feedback. We have carefully addressed your comments and have revised our paper accordingly. If you have any further questions, we would be eager to engage in further discussion with you.
> > > > >
> > > > > Additionally, we would like to take this opportunity to extend our warmest wishes for a joyful and restful Thanksgiving holiday to you and your team.
> > > > >
> > > > > Best regards,
> > > > >
> > > > > Authors of Submission 6007

---

> > > > > > ### Comment · Reviewer_duFd · 2024-12-02
> > > > > >
> > > > > > Response to A10, A11, A12
> > > > > >
> > > > > > **Response to A10**
> > > > > > - SeeSR has employed a cross-attention strategy in the control part, which aligns closely with ClearSR.
> > > > > > - Based on the results provided in tables, the trade-off between perception and fidelity is  influenced by changes to the window size. This trade-off can also be adjusted through various other tricks, such as modifying the CFG value or employing techniques like the restoration-guided sampling strategy proposed by SUPIR.
> > > > > >
> > > > > > Given these observations, I remain of the opinion that the technical contributions of ClearSR are limited.
> > > > > >
> > > > > > **Response to A11**
> > > > > >
> > > > > > SeeSR utilizes **LR representation embedding and tag-style text embedding** to enhance generative capabilities. This is reasonable, as both embeddings align with the text embedding space of the pre-trained T2I model.
> > > > > >
> > > > > > In contrast, ClearSR directly employs the **LR latent encoded by the VAE encoder** as the control signal. Intuitively, it is difficult to argue that this approach can effectively enhance generative capabilities.
> > > > > >
> > > > > > **Response to A12**
> > > > > >
> > > > > > In the early stages of diffusion, ClearSR adopted SupIR's strategy by weighting LR latent features to suppress structural errors. However, SupIR uses a progressive weighting strategy, which is more flexible than ClearSR's fixed value approach.
> > > > > >
> > > > > > In the later stages of diffusion, I agree with ClearSR's approach—emphasizing the generation of details based on faithful structures.
> > > > > >
> > > > > > **Decision**
> > > > > >
> > > > > > Based on the above analysis, I decide to maintain the initial score.

---

> > > > > > > ### Author Response · Authors · 2024-12-02
> > > > > > >
> > > > > > > Thank you for your valuable feedback. Our response is as follows:
> > > > > > >
> > > > > > > **Q13: SeeSR has employed a cross-attention strategy in the control part, which aligns closely with ClearSR. Based on the results provided in tables, the trade-off between perception and fidelity is influenced by changes to the window size. This trade-off can also be adjusted through various other tricks, such as modifying the CFG value or employing techniques like the restoration-guided sampling strategy proposed by SUPIR. Given these observations, I remain of the opinion that the technical contributions of ClearSR are limited.**
> > > > > > >
> > > > > > > A13: Thank you for your feedback. First, we need to clarify that **cross-attention is a widely adopted technique** used to integrate additional information into a module. This technique has been extensively applied across various tasks in computer vision, with many notable works utilizing it (e.g., DETR, ControlNet, SD3, etc.). Furthermore, in previous SD-based methods, cross-attention is also commonly employed (e.g., PASD, SeeSR, CoSeR, SUPIR, etc.).
> > > > > > >
> > > > > > > It is important to emphasize that **the main contribution of our method is the effective utilization of LR latent embeddings**. We leverage the widely adopted cross-attention mechanism to better integrate LR latent embeddings into the latent space. However, **cross-attention itself is not a core component of our method**. Similarly, SeeSR employs cross-attention to incorporate semantic signals, but its primary emphasis is on the significance of semantic signals for RealISR tasks. Moreover, we have detailed the technical advancements of our method in A10.
> > > > > > >
> > > > > > > Regarding the window size, in A10, we demonstrated that the window size can be used to balance fidelity and generation. However, the purpose of providing this ablation study is only to demonstrate the impact of the window size, and the effectiveness of the window partition is shown in Table 2 of the paper. **We mainly use the LSA strategy to achieve the balance between fidelity and generation instead of adjusting the window size**.
> > > > > > >
> > > > > > > **Q14: SeeSR utilizes LR representation embedding and tag-style text embedding to enhance generative capabilities. This is reasonable, as both embeddings align with the text embedding space of the pre-trained T2I model. In contrast, ClearSR directly employs the LR latent encoded by the VAE encoder as the control signal. Intuitively, it is difficult to argue that this approach can effectively enhance generative capabilities.**
> > > > > > >
> > > > > > > A14: Thank you for your feedback. We also need to clarify some concepts first.
> > > > > > >
> > > > > > > The LR representation embedding and tag-style text embedding used in SeeSR both come from RAM and are semantic signals. In the control branch and UNet, SeeSR integrates the semantic signals through cross-attention. However, the control branch of SeeSR is also based on ControlNet, where the input to this module is the LR latent embeddings output by an encoder (trained from scratch). Overall, for SeeSR, the final control signal is the result of integrating the semantic signals into the control branch based on ControlNet.
> > > > > > >
> > > > > > > Therefore, SeeSR designs an encoder and trains it from scratch to obtain the LR latent embedding and uses RAM to provide the LR representation embedding and tag-style text embedding (semantic signals). ClearSR, on the other hand, fine-tunes the pre-trained VAE from SD2.1 and uses the LR latent embedding to further constrain the features in the control branch.
> > > > > > >
> > > > > > > From the clarification above, it can be seen that the VAE used in ClearSR is more aligned with the pre-trained SD2.1 compared to an encoder trained from scratch. Additionally, the semantic signals provided by SeeSR come from RAM, which are not aligned with the latent space of SD.
> > > > > > >
> > > > > > > **Q15: In the early stages of diffusion, ClearSR adopted SupIR's strategy by weighting LR latent features to suppress structural errors. However, SupIR uses a progressive weighting strategy, which is more flexible than ClearSR's fixed value approach. In the later stages of diffusion, I agree with ClearSR's approach—emphasizing the generation of details based on faithful structures.**
> > > > > > >
> > > > > > > A15: Thank you for your feedback. We also need to emphasize that our LSA is based on our observations during the inference stage, and we have demonstrated the effectiveness of dividing the inference process into three parts in A12.
> > > > > > >
> > > > > > > Previous methods, including SUPIR, simply add the LR latent to the predicted $\mathbf{x_0}$ in different ways, without conducting an in-depth study of the inference stage.
> > > > > > >
> > > > > > > Based on our observations, our LSA adjusts the predicted $\mathbf{x_0}$ within a more reasonable range of inference steps, which is a more effective adjustment strategy. Our ELA only requires fixed weighting within the appropriate inference steps to achieve good results, without the need to design complex weighting strategies. Although in LSA, we can also use smaller weights closer to the Content Generation stage, this is not the core of our method.

---

### Official Review · Reviewer_4hYN · 2024-10-31

**Soundness:** 3
**Presentation:** 3
**Contribution:** 3
**Rating:** 5
**Confidence:** 5

**Summary:**

This paper introduces ClearSR, a novel approach designed to enhance the utilization of LR image information in SR tasks. The DPM and SPM modules are designed, enabling the extraction of more LR details and structural information. The method also demonstrates that latent LR embeddings can be used to adjust the latent space during inference, improving both fidelity and generative quality. ClearSR outperforms existing SR models across multiple metrics on various test datasets, producing SR results with rich generated details while maintaining consistency with the LR images.

**Strengths:**

1. This paper proposes two modules to extract more LR details for structural and detail preservation.

2. In the inference stage, this paper proposes an LSA strategy, which performs different directional adjustments towards LR embeddings in the latent space in the earlier and later steps. This idea is reasonable and interesting.

3. The results look good, the writing is well, and the paper is easy to follow.

**Weaknesses:**

1. This paper introduces two modules (DPM and SPM) to enhance the utilization of LR image information, but these increase model parameters and inference time compared to ControlNet. However, algorithmic complexity is not discussed.

2. The description in Line 190 is confusing; PASD does not use the CLIP image encoder to extract LR features.

3. The explanation of image-level feature $\textbf{p}$ in Figure 3 is unclear. How is $\textbf{p}$ integrated into SD Unet, and what is its role in the framework?

4. DPM and SPM are designed to extract LR information at detail and structure levels, both of which should contribute to fidelity. However, Table 2 suggests that SPM improves fidelity, while window-based cross-attention layers in DPM weaken fidelity. More explanation is required.

**Questions:**

1. The authors need to compare the complexity of ClearSR with that of the other methods, including the model parameter counts, inference time, and inference timestep.

2. The authors should double-check the understanding of PASD in Line 190.

3.  The authors should add a clearer description of the image-level feature $\textbf{p}$ in Figure 3. How is $\textbf{p}$ integrated into SD Unet, and what is its role in the framework?

4. The authors should explain more clearly in Table 2. Why does the SPM improve fidelity, while window-based cross-attention layers in DPM weaken fidelity? In addition, the ablation study that includes a model without DPM should also be provided for a more complete picture of each module's contribution.

---

> ### Author Response · Authors · 2024-11-21
>
> Thank you for your valuable feedback. Our response is as follows:
>
> **Q1: The authors need to compare the complexity of ClearSR with that of the other methods, including the model parameter counts, inference time, and inference timestep.**
>
> A1: Thank you for your thoughtful suggestion. We compare the complexity of our ClearSR with that of several SD-based Real-ISR methods (DiffBIR, PASD and SeeSR), including total parameters, trainable parameters, MACs, inference step, inference time and inference speed. All methods are tested on an A40 GPU. Although the additional layers increase the number of parameters and computational cost, we can see that our ClearSR has fewer total parameters, trainable parameters, and MACs compared to SeeSR. For inference speed, since the Diffusers library is optimized for the Classfier-Free Guidance (CFG), we disabled CFG during inference to achieve a fair comparison. Note that DiffBIR originally does not use CFG. In addition, we can also observe that our ClearSR performs well when the inference step is set to 20 (lower MACs and a reduced inference time). This further proves that our model has stronger generative capabilities, allowing it to recover good results even with fewer inference steps. We have added the complexity comparisons to Appendix E in the revision.
>
> | | DiffBIR | PASD | SeeSR | ClearSR | ClearSR-s20 |
> |-----------|-----------|-----------|-----------|-----------|-----------|
> | Total Param (M) | 1717 | 1900 | 2524 | 2511 | 2511 |
> | Trainable Param (M) | 380 | 625 | 750 | 525 | 525 |
> | MACs (G) | 24234 | 29125 | 65857 | 52384 | 21855 |
> | Inference Steps | 50 | 20 | 50 | 50 | 20 |
> | Inference Time (s) | 4.51 | 1.92 | 4.10 | 5.36 | 2.14 |
> | Inference Speed (step/s) | 11.09 | 10.41 | 12.21 | 9.33 | 9.33 |
>
> Quantitative comparison of ClearSR-s20 on DRealSR dataset is as below.
>
> | | PSNR $\uparrow$ | SSIM $\uparrow$ | LPIPS $\downarrow$ | NIQE $\downarrow$ | MUSIQ $\uparrow$ | MANIQA $\uparrow$ | CLIPIQA $\uparrow$ |
> |-----------|-----------|-----------|-----------|-----------|-----------|-----------|-----------|
> |PASD|27.36|0.7073|0.3760|5.5474|64.87|0.6169|0.6808|
> |SeeSR|28.17|0.7691|0.3189|6.3967|64.93|0.6042|0.6804|
> |ClearSR|28.22|0.7538|0.3473|6.0867|66.27|0.6246|0.6976|
> |ClearSR-s20|28.53|0.7689|0.3543|7.4823|65.88|0.6088|0.7176|
>
> **Q2: The authors should double-check the understanding of PASD in Line 190.**
>
> A2: Thank you for your reminder. In Line 190, “Similar to PASD (Yang et al., 2023), we let the LR image pass through the CLIP image encoder to obtain the image-level feature $\mathbf{p}$ and replace the null-text prompt in the UNet decoder”, which means the similar way to extract some high-level information using the pre-trained model. In contrast, PASD uses ResNet, YOLO, and BLIP to extract information, and then converts it into image-level features using the CLIP encoder. We directly use the CLIP encoder to process images, and then pass the feature output from the CLIP image encoder through two MLP layers to match the shape and adapt to the degradation of the LR. This feature is then used to replace the null-text prompt.
>
> **Q3: The authors should add a clearer description of the image-level feature $\mathbf{p}$ in Figure 3. How is $\mathbf{p}$ integrated into SD Unet, and what is its role in the framework?**
>
> A3: Thank you for your suggestion. As mentioned in A2, since $\mathbf{p}$ directly replaces the null-text prompt, it will interact with the UNet through the cross-attention layer, which originally interacted with the text embedding. In our framework, similar to PASD, $\mathbf{p}$ serves as additional high-level information to enhance the model's generative ability, but it may lead to a decrease in fidelity. We conduct the ablation study and the results are as follows:
>
> | | PSNR $\uparrow$ | SSIM $\uparrow$ | NIQE $\downarrow$ | MANIQA $\uparrow$ |
> |-----------|-----------|-----------|-----------|-----------|
> |ClearSR|27.73|0.7390|6.4525|0.6263|
> |ClearSR w/o $\mathbf{p}$|27.87|0.7626|6.7029|0.6193|

---

> ### Author Response · Authors · 2024-11-21
>
> **Q4: The authors should explain more clearly in Table 2. Why does the SPM improve fidelity, while window-based cross-attention layers in DPM weaken fidelity? In addition, the ablation study that includes a model without DPM should also be provided for a more complete picture of each module's contribution.**
>
> A4: Thank you for your thoughtful question. In Table 2, with the addition of window-based cross-attention layers, although PSNR shows a slight decrease, SSIM still shows improvement. The improvement of SSIM indicates that the structural information of the LR is better preserved. Considering PSNR and SSIM together, fidelity has not been weakened. On the other hand, the window-based cross-attention layers in DPM contribute to the improvement in generative metrics, demonstrating the effectiveness of our method.
> We also appreciate your suggestion to add an ablation study. It can be observed that without DPM, the model's fidelity decreases significantly, further demonstrating the effectiveness of our method.
>
> | | PSNR $\uparrow$ | SSIM $\uparrow$ | NIQE $\downarrow$ | MANIQA $\uparrow$ |
> |-----------|-----------|-----------|-----------|-----------|
> |ClearSR|27.62|0.7483|6.6334|0.6222|
> |ClearSR w/o DPM|26.88|0.7081|6.0081|0.6196|

---

> > ### Comment · Reviewer_4hYN · 2024-11-24
> >
> > Thank you for the authors’ response. While some concerns have been addressed, a few key points still require clarification.
> >
> > Regarding the response to Q2, there is a misunderstanding regarding using CLIP encoders. It’s important to distinguish between the CLIP text encoder and the CLIP image encoder, as conflating the two may confuse readers. Most diffusion-based methods, such as PASD and SeeSR, use the CLIP text encoder, not the CLIP image encoder. To highlight your use of the CLIP image encoder, I suggest citing CoSeR, which leverages it for extracting LR features.
> >
> > Regarding the response to Q4:
> > The paper claims that the proposed DPM and SPM can extract more LR information at both structural and detail levels, contributing to fidelity. However, ClearSR does not show a significant advantage in reference-based metrics (SSIM and LPIPS) over other diffusion-based methods in Table 1. This raises questions about the consistency between the problem the paper addresses and the presented results.
> >
> > In addition, why are the ClearSR results on DrealSR inconsistent across Tables of the Q3 and Q4 responses, Table 1, and Table 2 of the main paper?

---

> > > ### Author Response · Authors · 2024-11-25
> > >
> > > Thank you for your reply. Regarding your point, here is our further discussion:
> > >
> > > **Q5: Regarding the response to Q2, there is a misunderstanding regarding using CLIP encoders. It’s important to distinguish between the CLIP text encoder and the CLIP image encoder, as conflating the two may confuse readers. Most diffusion-based methods, such as PASD and SeeSR, use the CLIP text encoder, not the CLIP image encoder. To highlight your use of the CLIP image encoder, I suggest citing CoSeR, which leverages it for extracting LR features.**
> > >
> > > A5: Thank you for your reminder. We have revised the parts that could cause confusion for the readers and have cited CoSeR to help readers better understand our method.
> > >
> > > **Q6: Regarding the response to Q4: The paper claims that the proposed DPM and SPM can extract more LR information at both structural and detail levels, contributing to fidelity. However, ClearSR does not show a significant advantage in reference-based metrics (SSIM and LPIPS) over other diffusion-based methods in Table 1. This raises questions about the consistency between the problem the paper addresses and the presented results.**
> > >
> > > A6: Thank you for your valuable question. In Table 2 in the paper and our response to Q4, we have shown the effectiveness of DPM and SPM on fidelity. However, there are other factors that also influence fidelity, which we will discuss in detail below.
> > >
> > > Firstly, the choice of window size in the window-based cross-attention layers of DPM is one of the reasons why our model does not outperform SeeSR and StableSR in terms of SSIM and LPIPS. As we can see, increasing the window size leads to a decrease in fidelity, while decreasing the window size results in a reduction in the model’s generative ability. To balance fidelity and generation, we selected 16 as the window size for ClearSR.
> > >
> > > | window size| PSNR $\uparrow$ | SSIM $\uparrow$ | LPIPS $\downarrow$ | NIQE $\downarrow$ | MANIQA $\uparrow$ |
> > > |-----------|-----------|-----------|-----------|-----------|-----------|
> > > |32|27.29|0.7294|0.3769|6.5364|0.6333|
> > > |16|27.62|0.7483|0.3646|6.6334|0.6222|
> > > |8|27.97|0.7619|0.3520|6.9919|0.6090|
> > >
> > > Moreover, fidelity is also influenced by other settings such as Classifier-Free Guidance (CFG) scale, CFG prompt, etc. In ClearSR, the CFG scale is set to 7, the positive prompt is "clean, high-resolution, 8k, detailed, realistic", and the negative prompt is "dotted, noise, blur, lowres, smooth". In the table below, we present the impact of different CFG scales and prompts.
> > >
> > > The impact of the different CFG scales is as below:
> > >
> > > |CFG scale| PSNR $\uparrow$ | SSIM $\uparrow$ | LPIPS $\downarrow$ | NIQE $\downarrow$ | MANIQA $\uparrow$ |
> > > |-----------|-----------|-----------|-----------|-----------|-----------|
> > > |2.0| 28.95 | 0.7721 | 0.3278 | 6.8862 | 0.5767|
> > > |3.0| 28.83 | 0.7695 | 0.3292 | 6.6302 | 0.5887|
> > > |4.0| 28.69 | 0.7661 | 0.3321 | 6.4985 | 0.5989|
> > > |5.0| 28.54 | 0.7622 | 0.3381 | 6.3750 | 0.6083|
> > > |6.0| 28.37 | 0.7580 | 0.3417 | 6.2467 | 0.6175|
> > > |7.0| 28.22 | 0.7538 | 0.3473 | 6.0867 | 0.6246|
> > > |8.0| 28.07 | 0.7494 | 0.3528 | 6.0229 | 0.6308|
> > > |9.0| 27.91 | 0.7451 | 0.3584 | 6.0677 | 0.6373|
> > > |10.0|27.77| 0.7408 | 0.3640 | 6.0263 | 0.6429|
> > >
> > > Different CFG prompts are as follows:
> > >
> > > P1: The positive prompt is "clean, high-resolution, 8k", and the negative prompt is "dotted, noise, blur, lowres, smooth".
> > >
> > > P2: The positive prompt is "continuous, clean, sharp, highres, textureddotted, noise, blur, lowres, smooth", and the negative prompt is "".
> > >
> > > P3: The positive prompt is "Cinematic, High Contrast, highly detailed, taken using a Canon EOS R camera, hyper detailed photo - realistic maximum detail, 32k, Color Grading, ultra HD, extreme meticulous detailing, skin pore detailing, hyper sharpness, perfect without deformationsy", and the negative prompt is "painting, oil painting, illustration, drawing, art, sketch, oil painting, cartoon, CG Style, 3D render, unreal engine, blurring, dirty, messy, worst quality, low quality, frames, watermark, signature, jpeg artifacts, deformed, lowres, over-smooth".
> > >
> > > P4: The positive prompt is "high quality, clean, sharp, highres, textured", and the negative prompt is "low quality, blurry, unsharp, low-resolution, weird textures".
> > >
> > > P5 (ClearSR): The positive prompt is "clean, high-resolution, 8k, detailed, realistic", and the negative prompt is "dotted, noise, blur, lowres, smooth".
> > >
> > > The impact of the different CFG prompts is as below:
> > >
> > > |Prompt| PSNR $\uparrow$ | SSIM $\uparrow$ | LPIPS $\downarrow$ | NIQE $\downarrow$ | MANIQA $\uparrow$ |
> > > |-----------|-----------|-----------|-----------|-----------|-----------|
> > > |P1|28.13|0.7532|0.3498|6.3232|0.6299|
> > > |P2|28.16|0.7506|0.3454|6.3499|0.6158|
> > > |P3|27.21|0.7041|0.3913|6.8144|0.6253|
> > > |P4|28.70|0.7532|0.3551|7.1183|0.5868|
> > > |P5|28.22|0.7538|0.3473|6.0867|0.6246|

---

> > > ### Author Response · Authors · 2024-11-25
> > >
> > > As can be seen, window size, CFG scale, and CFG prompt all have an impact on the fidelity. To strike a balance between fidelity and generation, we ultimately chose the appropriate settings for the final version in Table 1. Moreover, we also hope our method has strong generative capability. As shown in Figure 1, our model can produce outputs more consistent with LR images with properly generated details. Therefore, our results are not inconsistent with our method.
> > >
> > > **Q7: Why are the ClearSR results on DrealSR inconsistent across Tables of the Q3 and Q4 responses, Table 1, and Table 2 of the main paper?**
> > >
> > > A7: Apologies for the confusion caused by these results. As mentioned in Section 4.1 and Section 4.3 in the paper, the ClearSR in Table 1 was trained for **150K** steps with a learning rate of $5 \times 10^{-5}$. The models in Table 2 and other ablation studies were trained for **50K** steps with a learning rate of $1 \times 10^{-4}$.
> > >
> > > In this response, our model was trained for **50K** steps with a learning rate of $5 \times 10^{-5}$. Due to time constraints, the response to Q3 is an exception, as it uses a historical experiment result where the model was trained for **110K** steps with a learning rate of $1 \times 10^{-4}$.
> > >
> > > However, these settings all allow the model to converge properly. The different settings in the ablation studies are due to computational limitations and time constraints and do not influence the experimental outcomes.
> > >
> > > If our explanation still confuses you, please feel free to submit a new response to us anytime.

---

> > > ### Author Response · Authors · 2024-12-01
> > >
> > > Dear Reviewer 4hYN,
> > >
> > > Thank you once again for your valuable feedback. We have carefully addressed your comments and have revised our paper accordingly. If you have any further questions, we would be eager to engage in further discussion with you.
> > >
> > > Additionally, we would like to take this opportunity to extend our warmest wishes for a joyful and restful Thanksgiving holiday to you and your team.
> > >
> > > Best regards,
> > >
> > > Authors of Submission 6007

---

> > > > ### Comment · Reviewer_4hYN · 2024-12-02
> > > >
> > > > Thank you for the authors’ response. However, some of my concerns remain unresolved.
> > > >
> > > > The paper's claims and results appear inconsistent. Table 1 shows that ClearSR does not exhibit a significant advantage in reference-based metrics (SSIM and LPIPS) compared to other diffusion-based methods, despite claiming that DPM and SPM enhance fidelity by extracting more LR information. The response demonstrates that adjusting settings (e.g., window size) can yield different fidelity-perception trade-offs, which suggests the proposed strategies do not effectively focus on extracting LR information to enhance fidelity. In addition, I suggest adding a clear explanation in the paper about the reasons why changing these settings affects the results.
> > > >
> > > > The ablation study results, obtained with an insufficient training process, are unconvincing. Since convergence speeds vary across settings, I suggest using the same training process as the main experiment. However, given time constraints, re-performing all experiments may not be feasible.
> > > >
> > > > Based on these issues, I would like to give a borderline score.

---

> > > > > ### Author Response · Authors · 2024-12-02
> > > > >
> > > > > Thank you for your valuable feedback. Our response is as follows:
> > > > >
> > > > > It is important to emphasize that the main contribution of our method is on the efficient use of LR latent embeddings. Although ablation studies have shown that both SPM and DPM are beneficial for fidelity, we still aim for the model to have strong generative capabilities. (As shown in Figure 1, the results generated by our model are more consistent with the LR image, but the trees contain more details.)
> > > > >
> > > > > - Regarding SSIM and LPIPS: In Table 1, our ClearSR does not show a significant advantage over other methods in terms of SSIM and LPIPS. However, our ClearSR* demonstrates a certain advantage in SSIM compared to SD-based methods and shows competitive performance in LPIPS, reflecting the effectiveness of our fidelity-perception trade-off method. Additionally, for LPIPS, it is primarily related to the texture in the results. Outputs with finer textures tend to have a worse LPIPS score. In the table below, we can see that for models with stronger generative capabilities like SUPIR, LPIPS is much lower. Therefore, combined with the advantages demonstrated in PSNR, NIQE, MUSIQ, MANIQA, and CLIPIQA, our results demonstrated the effectiveness of our approach.
> > > > >
> > > > > | | PSNR $\uparrow$ | SSIM $\uparrow$ | LPIPS $\downarrow$ | NIQE $\downarrow$ | MUSIQ $\uparrow$ | MANIQA $\uparrow$ | CLIPIQA $\uparrow$ |
> > > > > |-----------|-----------|-----------|-----------|-----------|-----------|-----------|-----------|
> > > > > |PASD|27.36|0.7073|0.3760|5.5474|64.87|0.6169|0.6808|
> > > > > |SeeSR|28.17|0.7691|0.3189|6.3967|64.93|0.6042|0.6804|
> > > > > |SUPIR|24.91|0.6348|0.4338|7.2245|60.43|0.5565|0.6887|
> > > > > |ClearSR|28.22|0.7538|0.3473|6.0867|66.27|0.6246|0.6976|
> > > > >
> > > > >
> > > > > - Regarding DPM: Although we have presented an ablation study on window size in A6, the purpose of providing this ablation study is only to demonstrate the impact of the window size. We mainly use the LSA strategy to achieve the balance between fidelity and generation instead of adjusting the window size. The effectiveness of DPM is shown in A3. We appreciate your suggestion, and we will add explanations about these settings in the paper.
> > > > >
> > > > > Thank you again for your valuable work. Your suggestions have greatly improved the quality of our paper.

---

### Official Review · Reviewer_wnGp · 2024-11-01

**Soundness:** 3
**Presentation:** 3
**Contribution:** 3
**Rating:** 6
**Confidence:** 5

**Summary:**

This paper proposes a prior-based controlnet-like approach for image super-resolution. The motivation is to refine the conditional feature to improve the fidelity of the SR output while avoid the obvious degradation of generation ability. The proposed approach aims to achieve this goal from both the architecture design by introducing additional modules as well as cross-attention layers and the inference strategy by introducing proper guidance at difference inference steps. There are also some observations to support the design.

**Strengths:**

+ The motivation is clear and there are also some observations to provide the insights for the design of the approach.
+ The evaluation shows reasonable improvement of the proposed approach.
+ The paper is easy to follow.

**Weaknesses:**

- The additional modules introduced in this paper may also increase the cost of training and inference. Some evaluation on the complexity should be provided.
- The proposed Latent Space Adjustment strategy is somewhat tricky. How to choose ideal hyperparameters can be tough and case-by-case. Moreover, when the degradation is severe, adding LR guidance into the inference may leads to blurry outputs.
-Some strong baselines are missing ,e.g., SUPIR.

**Questions:**

My main concerns are as follows:

1. The author claims that ControlNet cannot preserve the LR information well in Figure 2. Is it because that ControlNet adds noise to the LR conditional during training and inference? Does the proposed approach also follows this setting as ControlNet? The authors should explicitly state whether they follow the same noise addition process as ControlNet, and if not, to explain how their approach differs.

2. The additional modules introduced in this paper may also increase the cost of training and inference. Some evaluation on the complexity should be provided, e.g., parameters, flops and inference time. The authors may consider provide some numerical comparison with existing baselines.

3. The proposed Latent Space Adjustment strategy is somewhat tricky. How to choose ideal hyperparameters can be tough and case-by-case. Moreover, when the degradation is severe, adding LR guidance into the inference may leads to blurry outputs. The authors should consider providing guidelines or heuristics for choosing hyperparameters, and discussing how their method performs under severe degradation conditions and the quality of the guidance under such cases.

4. SUPIR has more powerful generative ability then the baselines in the paper. The authors may want to explain why SUPIR was not included as a baseline, or to consider adding it to their comparisons if feasible.

5. Why choosing window cross-attention rather than full-attention and how to decide the window size? The authors should provide empirical or theoretical justification for using window cross-attention, and explain how they determined the optimal window size.

---

> ### Author Response · Authors · 2024-11-21
>
> Thank you for your valuable feedback. Our response is as follows:
>
> **Q1: The author claims that ControlNet cannot preserve the LR information well in Figure 2. Is it because that ControlNet adds noise to the LR conditional during training and inference? Does the proposed approach also follows this setting as ControlNet? The authors should explicitly state whether they follow the same noise addition process as ControlNet, and if not, to explain how their approach differs.**
>
> A1: Thank you for your question. Firstly, according to the DiffBIR paper, there is an ablation study on adding noise to the LR condition. Using only the LR condition as input to the control branch results in improved fidelity but a decrease in generative metrics. However, in this experiment, the changes in both fidelity and generative metrics are quite slight, indicating that noise has a negligible impact on the control branch. This experiment demonstrates that noise is not the primary reason why ControlNet fails to preserve the LR information well.
> Additionally, adding noise to the LR condition does introduce some randomness. However, in DiffBIR, the noise is concatenated with the LR condition, which means that the impact of the noise is reduced. In ClearSR, our code is based on DiffBIR, so the noise addition process is the same as in DiffBIR. Furthermore, the noise addition process for ControlNet in Figure 2 of the ClearSR paper also employs the concatenation approach. We can see that even with the concatenation approach, the final control signal still cannot preserve the LR information well.
>
> **Q2: The additional modules introduced in this paper may also increase the cost of training and inference. Some evaluation on the complexity should be provided, e.g., parameters, flops and inference time. The authors may consider provide some numerical comparison with existing baselines.**
>
> A2: Thank you for your thoughtful suggestion. We compare the complexity of our ClearSR with that of several SD-based Real-ISR methods (DiffBIR, PASD and SeeSR), including total parameters, trainable parameters, MACs, inference step, inference time and inference speed. All methods are tested on an A40 GPU. Although the additional layers increase the number of parameters and computational cost, we can see that our ClearSR has fewer total parameters, trainable parameters, and MACs compared to SeeSR. For inference speed, since the Diffusers library is optimized for the Classfier-Free Guidance (CFG), we disabled CFG during inference to achieve a fair comparison. Note that DiffBIR originally does not use CFG. In addition, we can also observe that our ClearSR performs well when the inference step is set to 20 (lower MACs and a reduced inference time). This further proves that our model has stronger generative capabilities, allowing it to recover good results even with fewer inference steps. We have added the complexity comparisons to Appendix E in the revision.
>
> | | DiffBIR | PASD | SeeSR | ClearSR | ClearSR-s20 |
> |-----------|-----------|-----------|-----------|-----------|-----------|
> | Total Param (M) | 1717 | 1900 | 2524 | 2511 | 2511 |
> | Trainable Param (M) | 380 | 625 | 750 | 525 | 525 |
> | MACs (G) | 24234 | 29125 | 65857 | 52384 | 21855 |
> | Inference Steps | 50 | 20 | 50 | 50 | 20 |
> | Inference Time (s) | 4.51 | 1.92 | 4.10 | 5.36 | 2.14 |
> | Inference Speed (step/s) | 11.09 | 10.41 | 12.21 | 9.33 | 9.33 |
>
> Quantitative comparison of ClearSR-s20 on DRealSR dataset is as below.
>
> | | PSNR $\uparrow$ | SSIM $\uparrow$ | LPIPS $\downarrow$ | NIQE $\downarrow$ | MUSIQ $\uparrow$ | MANIQA $\uparrow$ | CLIPIQA $\uparrow$ |
> |-----------|-----------|-----------|-----------|-----------|-----------|-----------|-----------|
> |PASD|27.36|0.7073|0.3760|5.5474|64.87|0.6169|0.6808|
> |SeeSR|28.17|0.7691|0.3189|6.3967|64.93|0.6042|0.6804|
> |ClearSR|28.22|0.7538|0.3473|6.0867|66.27|0.6246|0.6976|
> |ClearSR-s20|28.53|0.7689|0.3543|7.4823|65.88|0.6088|0.7176|

---

> ### Author Response · Authors · 2024-11-21
>
> **Q3: The proposed Latent Space Adjustment strategy is somewhat tricky. How to choose ideal hyperparameters can be tough and case-by-case. Moreover, when the degradation is severe, adding LR guidance into the inference may leads to blurry outputs. The authors should consider providing guidelines or heuristics for choosing hyperparameters, and discussing how their method performs under severe degradation conditions and the quality of the guidance under such cases.**
>
> A3: Thank you for your valuable question.
> We will first answer the question about degradation and then discuss how to choose hyperparameters.
> Regarding the question of how ClearSR performs under severe degradation conditions, firstly, we used LoRA layers to fine-tune the VAE, enabling our model to adapt to severe degradation conditions. Please note that during inference, we also use the fine-tuned VAE to provide LR guidance.
> We conducted the experiment on the DRealSR dataset, where we added extra degradation to the LR images to simulate severe degradation conditions. Using the HR image as the reference, we calculated the PSNR, SSIM, and LPIPS before and after adding degradation:
>
> || PSNR $\uparrow$ | SSIM $\uparrow$| LPIPS$ \downarrow$ |
> |-----------|-----------|-----------|-----------|
> |Before adding degradations|30.57|0.8301|0.4608|
> |After adding degradations|29.03|0.7961|0.5698|
>
> Subsequently, we input the degraded images into SeeSR and ClearSR. The table below shows the metrics before and after adding degradation:
>
> || PSNR $\uparrow$ | SSIM $\uparrow$ | NIQE $\downarrow$ | MANIQA $\uparrow$ |
> |-----------|-----------|-----------|-----------|-----------|
> |SeeSR before|28.17|0.7691|6.3967|0.6042|
> |SeeSR after|27.47|0.7525|6.6463|0.6028|
> |ClearSR before|28.22|0.7538|6.0867|0.6246|
> |ClearSR after|27.74|0.7391|6.3062|0.6221|
>
> We further calculated the changes in these metrics before and after adding degradation:
>
> | | Δ PSNR $\uparrow$ | Δ SSIM $\uparrow$ | Δ NIQE $\downarrow$ | Δ MANIQA $\uparrow$ |
> |-----------|-----------|-----------|-----------|-----------|
> |SeeSR|-0.70|-0.0166|+0.2496|-0.0014|
> |ClearSR|-0.48|-0.0147|+0.2195|-0.0025|
>
> We can see that the generative metric changes for SeeSR and ClearSR are relatively small. However, ClearSR shows a smaller decrease in PSNR and SSIM, indicating that it adapts better to severe degradation conditions.
> Next, we demonstrate the impact of different hyperparameters of LSA under severe degradation conditions, and compare these results with those under normal degradation conditions (before adding extra degradations). As seen, even in severe degradation conditions, increasing α still improves fidelity, and the effect is similar to that observed under normal degradation conditions. Therefore, LSA performs well under severe degradation conditions, and adding LR guidance into the inference under high degradation conditions does not lead to blurry outputs.
>
> ||α|β| PSNR $\uparrow$ | SSIM $\uparrow$ | NIQE $\downarrow$ | MANIQA $\uparrow$ |
> |-----------|-----------|-----------|-----------|-----------|-----------|-----------|
> |ClearSR before|0.01|0.01|28.22|0.7538|6.0867|0.6246|
> |ClearSR before|0.02|0.01|28.62|0.7677|6.4747|0.6071|
> |ClearSR before|0.03|0.01|29.00|0.7781|6.9796|0.5878|
> |ClearSR after|0.01|0.01|27.74|0.7391|6.3062|0.6221|
> |ClearSR after|0.02|0.01|28.12|0.7545|6.7463|0.5986|
> |ClearSR after|0.03|0.01|28.48|0.7645|7.3075|0.5759|
>
> Then, regarding the choice of LSA hyperparameters:
> 1. Our LSA method is designed to balance fidelity and generation. However, the "optimal balance" between fidelity and generation is relatively subjective. In the paper, we tested multiple sets of α and β values and ultimately selected the ones that we believed were relatively well balanced in fidelity and generation.
> 2. As mentioned above, our ClearSR adapts well to severe degradation conditions. Using the default settings generally provides good results. Therefore, users do not need to select hyperparameters case-by-case. Moreover, LSA allows for a wide range of adjustments, so users can adjust the hyperparameters to their specific needs to achieve their desired results.
> 3. We can also use a simple strategy for automatic hyperparameter selection. For instance, by first calculating the MANIQA score of the LR image and then choosing different values of α and β based on the MANIQA score. Our strategy is demonstrated in the table below:
>
> |Group|MANIQA=m|α|β|
> |-----------|-----------|-----------|-----------|
> |1|m<0.35|0.000|0.015|
> |2|0.35≤m＜0.45|0.005|0.010|
> |3|0.45≤m<0.55|0.010|0.005|
> |4|0.55≤m|0.015|0.000|
>
> In the table below, we present the metrics using this strategy:
>
> || PSNR $\uparrow$ | SSIM $\uparrow$ | NIQE $\downarrow$ | MANIQA $\uparrow$ |
> |-----------|-----------|-----------|-----------|-----------|
> |ClearSR base|28.22|0.7538|6.0867|0.6246|
> |ClearSR auto|28.26|0.7552|6.2290|0.6221|

---

> ### Author Response · Authors · 2024-11-21
>
> Although our LSA method performs similarly under various degradation conditions, meaning this strategy may not improve overall metrics, it does have specific effects. We present the metrics for each group of LR images below. We can see that for lower-quality LR images (Group 1, Group 2), our model tends to generate more details (as shown by improvements in NIQE and MANIQA). For higher-quality LR images (Group 3, Group 4), the model outputs results more consistent with the LR image (as shown by improvements in PSNR and SSIM). Overall, using this strategy enhances the visual quality of the model's output.
>
> |Group|MANIQA=m| PSNR $\uparrow$ | SSIM $\uparrow$ | NIQE $\downarrow$ | MANIQA $\uparrow$ |
> |-----------|-----------|-----------|-----------|-----------|-----------|
> |1 base|m<0.35|25.82|0.7205|6.9769|0.6444|
> |1 auto|m<0.35|25.31|0.6919|6.5023|0.6681|
> |2 base|0.35≤m＜0.45|29.35|0.7989|6.8092|0.6034|
> |2 auto|0.35≤m＜0.45|29.15|0.7925|6.6800|0.6128|
> |3 base|0.45≤m<0.55|28.18|0.7431|5.6731|0.6257|
> |3 auto|0.45≤m<0.55|28.29|07469|5.8528|0.6222|
> |4 base|0.55≤m|26.69|0.7136|6.0585|0.6583|
> |4 auto|0.55≤m|27.03|0.7279|6.7394|0.6443|
>
> Additionally, automatically selecting hyperparameters is a complex task that typically requires extensive engineering investigation to determine the final solution. Therefore, we did not report these results in the paper. However, we will continue to conduct related research to further enhance the visual quality of our model's output.
>
> **Q4: SUPIR has more powerful generative ability than the baselines in the paper. The authors may want to explain why SUPIR was not included as a baseline, or to consider adding it to their comparisons if feasible.**
>
> A4: Thank you for your suggestion. Although SUPIR demonstrates stronger generative capabilities, the comparison is unfair due to differences in the dataset and model size. We will explain these two aspects in detail.
>
> In terms of datasets, SUPIR collected a private dataset of 20 million high-resolution, high-quality images for model training. However, ClearSR uses the same datasets as SeeSR and PASD, which consist of DIV2K, Flickr2K, DIV8K, OST, and the first 10K face images from FFHQ, totaling approximately 25,000 images. SUPIR's dataset is 800 times larger.
>
> Regarding the model, SUPIR is based on SDXL. However, ClearSR is based on SD2.1-base and is only one-third the size of SDXL. Other baselines also typically use models of similar size, such as SeeSR using SD2-base and StableSR using SD2.1-base.
> Overall, comparisons with SUPIR are unfair in terms of both datasets and model size, and the dataset used by SUPIR is difficult to obtain. Therefore, we did not include SUPIR as a baseline but discussed it in the paper.
>
> **Q5: Why choosing window cross-attention rather than full-attention and how to decide the window size? The authors should provide empirical or theoretical justification for using window cross-attention, and explain how they determined the optimal window size.**
>
> A5: Thank you for your question. In Table 2, we demonstrate that not using window partition results in a decrease in fidelity. Additionally, we found that smaller window size improves fidelity but results in a reduction in generative ability. The results of the ablation study on window size are as follows:
>
> | window size| PSNR $\uparrow$ | SSIM $\uparrow$ | NIQE $\downarrow$ | MANIQA $\uparrow$ |
> |-----------|-----------|-----------|-----------|-----------|
> |32|27.29|0.7294|6.5364|0.6333|
> |16|27.62|0.7483|6.6334|0.6222|
> |8|27.97|0.7619|6.9919|0.6090|
>
> We can see that increasing the window size leads to a decrease in fidelity, while decreasing the window size results in a reduction in the model's generative ability. To strike a balance between fidelity and generative ability, we finally chose a window size of 16 for ClearSR. We have added this ablation study to Appendix F in the revision.

---

> > ### Comment · Reviewer_wnGp · 2024-11-23
> > **Official Comment by Reviewer wnGp**
> >
> > The author addresses most of my concerns.
> >
> > While the author mentioned the unfairness compared with SUPIR, I still think it is necessary to compare this SOTA approach. After all, SR is mainly aimed at real-world applications, and it is not reasonable to ignore the approach with SOTA performance. Besides, it is also hard to say it is fair to compare with other baselines. For example, ClearSR has more than 800M parameters than DiffBIR. Thus, the author is suggested to make a comparison and explain the gap which is understandable. Also, the author may consider applying the proposed strategies on larger models such as SUPIR in the future to make sure that the proposed approach can generally work in large models.
> >
> > On the other hand, I also agree with Reviewer duFd in terms of the novelty issues and the author should make further comparisons with existing approaches.

---

> > > ### Author Response · Authors · 2024-11-25
> > >
> > > Thank you for your reply. We are glad we could address most of your concerns.
> > >
> > > **Q6: While the author mentioned the unfairness compared with SUPIR, I still think it is necessary to compare this SOTA approach. After all, SR is mainly aimed at real-world applications, and it is not reasonable to ignore the approach with SOTA performance. Besides, it is also hard to say it is fair to compare with other baselines. For example, ClearSR has more than 800M parameters than DiffBIR. Thus, the author is suggested to make a comparison and explain the gap which is understandable. Also, the author may consider applying the proposed strategies on larger models such as SUPIR in the future to make sure that the proposed approach can generally work in large models.**
> > >
> > > A6: Thank you for your thoughtful suggestion. We will first provide a quantitative comparison with SUPIR.
> > >
> > > We test on the RealPhoto60 proposed by SUPIR and DRealSR test sets. We used the default settings of SUPIR for testing. The table below shows the results for RealPhoto60:
> > >
> > > || NIQE $\downarrow$ | MUSIQ $\uparrow$ | MANIQA $\uparrow$ |CLIPIQA $\uparrow$ |
> > > |-----------|-----------|-----------|-----------|-----------|
> > > |SUPIR|3.2494|70.4547|0.6467|0.6983|
> > > |ClearSR|3.8839|68.2425|0.6203|0.6281|
> > >
> > > As can be seen, SUPIR demonstrates stronger generative capabilities. SUPIR can generate impressive details, especially good at generating textures of trees, flowers, and plants.
> > >
> > > The table below shows the results for DRealSR:
> > >
> > > | | PSNR $\uparrow$ | SSIM $\uparrow$ | LPIPS $\downarrow$ | NIQE $\downarrow$ | MUSIQ $\uparrow$ | MANIQA $\uparrow$ | CLIPIQA $\uparrow$ |
> > > |-----------|-----------|-----------|-----------|-----------|-----------|-----------|-----------|
> > > |PASD|27.36|0.7073|0.3760|5.5474|64.87|0.6169|0.6808|
> > > |SeeSR|28.17|0.7691|0.3189|6.3967|64.93|0.6042|0.6804|
> > > |SUPIR|24.91|0.6348|0.4338|7.2245|60.43|0.5565|0.6887|
> > > |ClearSR|28.22|0.7538|0.3473|6.0867|66.27|0.6246|0.6976|
> > >
> > > As can be seen, due to SUPIR's strong generative capabilities, the fidelity decreases, which is consistent with the description in the SUPIR paper. Actually, in the DRealSR test set, SUPIR's outputs exhibit two extremes: some images generate very rich details, while others produce many fake textures. The former has a negative impact on fidelity, while the latter has a negative impact on generative metrics.
> > >
> > > Moerover, both of these cases show that while SUPIR has strong generative capabilities, the overly strong generative ability of the pre-trained SDXL might lead to inconsistent super-resolution results. In our ClearSR, we focus more on ensuring that outputs are more consistent with the LR images while maintaining generative capability. This good constraint of the generative prior also reflects the potential of our methods when applied to larger models (such as SDXL).
> > >
> > > We have uploaded some of the test results in supplementary materials. If you have any further questions, please feel free to submit a new response to us anytime.

---

> > > ### Author Response · Authors · 2024-11-25
> > >
> > > **Q7: The author should make further comparisons with existing approaches.**
> > >
> > > A7: Thank you for your suggestion. We will make further comparisons with existing approaches as below.
> > >
> > > From the perspective of **control signal optimization**, our design principles are fundamentally different from existing methods (such as PASD and SeeSR).
> > >
> > > PASD introduces the PACA, which allows the control signal, before passing through the zero convolution layer, to directly interact with the features in the UNet. The goal of this design is to better integrate the control signal into the UNet. **It is essentially an efficient use of the control signal, but it does not improve the control signal itself**. In contrast, as shown in Figure 2 of the ClearSR paper, we observed that the control signal provided by the original ControlNet has a bias relative to the LR latent embedding. Strengthening the utilization of such biased control signals still cannot provide accurate guidance to the UNet, which limits the model's potential.
> > >
> > > SeeSR uses semantic information to enhance the model's generative capability, **essentially leveraging the priors of the fine-tuned RAM to provide additional information to the control signal. However, SeeSR does not address the issue of consistency between the model's output and the LR image**. The semantic information obtained from the RAM is inherently biased, and as shown in Figure 1 of the ClearSR paper, this semantic signal might lead to inconsistent generation with the LR image. This means that while SeeSR enriches the information contained in the control signal, it might have a negative impact due to the bias in the semantic signal.
> > >
> > > **Our design principle is that only when the quality of the control signal itself is improved will the model reach a higher potential, which represents a new paradigm in this field**. From this perspective, we apply constraints on the control signal in the latent space. We optimize the control signal from both detail and structural level. Since all the information comes solely from the LR latent embeddings, this ensures that the model's generation is not guided in the wrong direction.
> > >
> > > Additionally, as mentioned in our response to Q7, in larger models, powerful generative priors tend to encourage the model to generate more details. In this term, our **ClearSR provides a solution for constraining these overly strong generative priors**.
> > >
> > > ---
> > >
> > > Regarding our proposed **LSA**, we will compare it in detail with the inference strategies of PASD, SUPIR, and DiffBIR.
> > >
> > > Firstly, as shown in Table 4 of the paper, Early-step LR Adjustment (ELA) could improve the fidelity, and Later-step LR Adjustment (LLA) could improve the generative capability, respectively. This means that using LSA allows the model to have stronger generative capabilities compared to the base model. Moreover, our strategy can **improve the fidelity and generation simultaneously** through appropriate settings. Since PASD, SUPIR, and DiffBIR methods can only improve fidelity unidirectionally, these are our unique advantages.
> > >
> > > **Comparison with PASD**: PASD only studied adjustments for the early stage. PASD attempts to eliminate the inconsistency in residual signals between training and testing by adding LR latent to the initial Gaussian noise.
> > >
> > > In contrast, we divide the entire inference stage into three parts and apply different adjustment strategies for the early and later steps. Additionally, our utilization of LR latent also differs from PASD. We directly adjust the predicted $\mathbf{x_0}$ in the latent space.
> > >
> > > **Comparison with SUPIR**: SUPIR applies the same adjustment strategy throughout the entire inference stage. However, in later steps，SUPIR still aims for each prediction in the later steps to **be closer to** the LR latent. However, based on our division of the inference stage, in the later steps, the model mainly focuses on detail enhancement.
> > >
> > > In contrast to SUPIR, we leverage the inherent properties of the LR image, which mainly contains structural information and has fewer details compared to HR images, to adjust the predicted $\mathbf{x_0}$ in the opposite direction, which means let $\mathbf{x_0}$ **move away from** the LR latent. Essentially, this provides guidance to the model, allowing it to generate more details in the appropriate regions. This guidance allows our LLA to improve the model's generative capabilities.

---

> > > ### Author Response · Authors · 2024-11-25
> > >
> > > **Comparison with DiffBIR**: DiffBIR optimizes MSEGuidance by using sobel operators to compute the gradient magnitude. This can provide region-adaptive guidance to the model, allowing the model to generate more details in high-frequency regions while to generate fewer details in low-frequency regions. It is important to note that the goal of this strategy is still to bring the predicted $\mathbf{x_0}$ closer to the LR latent, meaning it can only improve the model's fidelity. Furthermore, this MSEGuidance-based strategy needs additional computational cost.
> > >
> > > Our LSA directly utilizes the inherent properties of the LR image, which mainly contains structural information and has fewer details compared to HR images, to guide the model. Our LLA also helps the model generate details in appropriate regions, but unlike DiffBIR, it is more direct and does not require additional computational cost.
> > >
> > > We hope our comparisons addresses your concerns. If you have any further questions, please feel free to submit a new response to us anytime.

---

> > > ### Author Response · Authors · 2024-12-01
> > >
> > > Dear Reviewer wnGp,
> > >
> > > Thank you once again for your valuable feedback. We have carefully addressed your comments and have revised our paper accordingly. If you have any further questions, we would be eager to engage in further discussion with you.
> > >
> > > Additionally, we would like to take this opportunity to extend our warmest wishes for a joyful and restful Thanksgiving holiday to you and your team.
> > >
> > > Best regards,
> > >
> > > Authors of Submission 6007

---

### Official Review · Reviewer_h8Y1 · 2024-11-02

**Soundness:** 3
**Presentation:** 3
**Contribution:** 2
**Rating:** 5
**Confidence:** 4

**Summary:**

This paper propose new diffusion-based method, named ClearSR, which can use the LR latent embedding to guide diffusion to generate better results. In particular, the author designs two modules to effectively use the information of LR embedding and propose a adjust strategy to balance the fidelity and detail of SR results.

**Strengths:**

This paper is clear in describing its contributions and methodology.
The author analyzed the relationship between image fidelity and model generation capabilities, and attempted to propose a solution strategy.
The experimental arrangement is relatively reasonable, and the ablation study can prove the effectiveness of the strategies proposed by the author.

**Weaknesses:**

Some descriptions in the paper may lead to confusion. The authors classify detail information as high-frequency information and structural information as low-frequency information. However, edges can also represent structure and are actually considered high-frequency information. The authors should use more appropriate terminology to avoid ambiguity.

To balance the fidelity and details of results, the author propose Latent Space Adjustment (LSA) strategy. However, the experimental results do not clearly demonstrate that the proposed method performs better in terms of fidelity (PSNR, SSIM, LPIPS, etc). In addition, similar approaches have also appeared in DiffBIR and PASD, and the author should provide a thorough comparison with the strategies proposed by these other methods.

**Questions:**

The motivation is clear. However, there are some concerns regarding the proposed approach. Specifically, the LR latent embedding, which is the output of the VAE encoder, has a size of 4x64x64, while the input image is 3x512x512. Compared to the original image, the LR embedding loses a significant amount of spatial information. Therefore, the LR latent embedding may not be suitable for supplementing detail and structural information.

Figure 2 shows that the proposed method has a low KL divergence value between the control signal and the low-resolution latent embedding. This suggests that the authors have introduced two modules to achieve a similar distribution between the LR latent embedding and the control signal. So why not use the LR latent embedding directly? Furthermore, from past work (DiffBIR, PASD, SeeSR), we know that the role of the control branch is primarily to remove degradation and bring it closer to the HR distribution. However, the method proposed by the authors results in the distribution of the control branch outputs being closer to the distribution of LR latent embedding, which is puzzling.

---

> ### Author Response · Authors · 2024-11-21
>
> Thank you for your valuable feedback. Our response is as follows:
>
> **Q1: Some descriptions in the paper may lead to confusion. The authors classify detail information as high-frequency information and structural information as low-frequency information. However, edges can also represent structure and are actually considered high-frequency information. The authors should use more appropriate terminology to avoid ambiguity.**
>
> A1: Thank you for your thoughtful suggestion. Edges can indeed be considered as high-frequency information. In ClearSR, the DPM is proposed to preserve detailed information in the LR image, while the SPM is proposed to maintain the global structural information information. To avoid ambiguity, we believe that the term "Global Structure Preserving Module" more accurately conveys our model design than the "Structure Preserving Module" (SPM). We have corrected the name of the module and modified any descriptions that may cause confusion according to the suggestion in the revision.

---

> ### Author Response · Authors · 2024-11-21
>
> **Q2: The experimental results do not clearly demonstrate that the proposed method performs better in terms of fidelity (PSNR, SSIM, LPIPS, etc). In addition, similar approaches have also appeared in DiffBIR and PASD, and the author should provide a thorough comparison with the strategies proposed by these other methods.**
>
> A2: Thank you for your question. In Table 1 of the original paper, to balance fidelity and generation, we selected appropriate α and β values, which allowed ClearSR to outperform previous diffusion-based methods (DiffBIR, PASD, SeeSR) in both fidelity and generation. We also adjusted the LSA settings and introduced another version of ClearSR, ClearSR*, which surpasses diffusion-based methods (no generative priors) in terms of fidelity (PSNR, SSIM, LPIPS) and is closer to the fidelity of GAN-based methods.
> To more intuitively demonstrate the effectiveness of our method, we adjust the LSA settings and provide two additional versions, including ClearSR-a (α=0.05, β=0.01) and ClearSR-b (α=0.02, β=0.01). In these versions, ClearSR-a is compared with Real-ESRGAN and ResShift, while ClearSR-b is compared with SeeSR. We can see that ClearSR-a outperforms Real-ESRGAN and ResShift in terms of PSNR and MANIQA. Similarly, ClearSR-b performs better than SeeSR in terms of PSNR and MANIQA, which demonstrates that our LSA could perform better in terms of fidelity with appropriate α and β values.
>
> | | PSNR $\uparrow$ | SSIM $\uparrow$ | LPIPS$\downarrow$ | MANIQA $\uparrow$ |
> |-----------|-----------|-----------|-----------|-----------|
> |Real-ESRGAN|28.64|0.8053|0.2847|0.4907|
> |ResShift|28.46|0.7673|0.4006|0.4586|
> |ClearSR-a|29.51|0.7958|0.3231|0.5358|
> |SeeSR|28.17|0.7691|0.3189|0.6042|
> |ClearSR-b|28.62|0.7677|0.3349|0.6071|
>
> We also provide a detailed comparison with the inference strategies proposed by DiffBIR and PASD. Firstly, the inference strategies proposed by DiffBIR and PASD can only enhance fidelity but cannot improve the generative capability. In contrast, our LSA method can enhance both fidelity and generation by adjusting α and β. We adjust the LSA settings and provide three additional versions, including ClearSR-c (α=0.01, β=0.00), ClearSR-d (α=0.00, β=0.01), and ClearSR-e (α=0.01, β=0.01). As shown in the table below, we can see that our LSA can enhance the fidelity by increasing α (ClearSR-c) and can also enhance the generative capability of the model by increasing β (ClearSR-d). With appropriate settings, both the fidelity and generative capability of the model can be enhanced simultaneously (ClearSR-e).
>
> | | PSNR $\uparrow$ | SSIM $\uparrow$ | NIQE$\downarrow$ | MANIQA $\uparrow$ |
> |-----------|-----------|-----------|-----------|-----------|
> |ClearSR w/o any inference strategy| 28.11| 0.7419|6.5289|0.6226|
> |ClearSR c| 28.44| 0.7609|6.4765|0.6172|
> |ClearSR-d| 27.85| 0.7412|5.9875|0.6360|
> |ClearSR-e| 28.22| 0.7538|6.0867|0.6246|
>
> Furthermore, we apply the *inference strategies* proposed by DiffBIR and PASD to ClearSR. We also adjust the LSA settings and provide two additional versions, including ClearSR-f (α=0.01, β=0.01) and ClearSR-g (α=0.015, β=0.01). Specifically, ClearSR-f compares ClearSR with the DiffBIR strategy, and ClearSR-g compares ClearSR with the PASD strategy. Note that, since DiffBIR's method is based on MSE Guidance, it introduces additional computational cost. As a result, we use the standard MSE Guidance in ClearSR-f to achieve a fair comparison. As can be observed, when the generative metrics are similar, our method achieves higher fidelity. Moreover, since LSA can enhance both fidelity and generation by adjusting α and β separately, our method offers broader adaptability, further proving the effectiveness of our method.
>
> | | PSNR $\uparrow$ | SSIM $\uparrow$ | NIQE$\downarrow$ | MANIQA $\uparrow$ |
> |-----------|-----------|-----------|-----------|-----------|
> |ClearSR w/o any inference strategy| 28.11| 0.7419|6.5289|0.6226|
> |ClearSR w DiffBIR strategy| 29.20| 0.7783|6.1992|0.5982|
> |ClearSR-f| 29.37| 0.7787|6.1229|0.5925|
> |ClearSR w PASD strategy| 28.39| 0.7594|6.3440|0.6149|
> |ClearSR-g| 28.42| 0.7610|6.2867|0.6151|

---

> ### Author Response · Authors · 2024-11-21
>
> **Q3: The LR latent embedding, which is the output of the VAE encoder, has a size of 4x64x64, while the input image is 3x512x512. Compared to the original image, the LR embedding loses a significant amount of spatial information. Therefore, the LR latent embedding may not be suitable for supplementing detail and structural information.**
>
> A3: Thank you for your valuable question. The LR embeddings do lose some information, but a well-trained VAE is still capable of preserving rich LR information. We use the VAE from SD 2.1-base to reconstruct images from the DRealSR test sets.
>
> || PSNR $\uparrow$ | SSIM $\uparrow$ | LPIPS $\downarrow$ |
> |-----------|-----------|-----------|-----------|
> |VAE reconstruction results| 39.78 | 0.9496 | 0.0335 |
> |Real-ESRGAN| 28.64 | 0.8053 | 0.2847|
> |SeeSR| 28.17 | 0.7691 | 0.3189|
> |ClearSR*| 29.00 | 0.7781 | 0.3281 |
>
> We can see that the reconstruction results show great PSNR, SSIM, and LPIPS scores. These results are much better than the output of existing methods, demonstrating the potential of LR embeddings. Additionally, how to integrate LR information into the SD model is a challenge, often requiring alignment between the input information and the latent space of SD. From this perspective, VAE is a useful tool. After fine-tuning VAE with the LoRA layers, it can well map LR images to a proper latent space while retaining rich information.
>
> **Q4: Figure 2 shows that the proposed method has a low KL divergence value between the control signal and the low-resolution latent embedding. This suggests that the authors have introduced two modules to achieve a similar distribution between the LR latent embedding and the control signal. So why not use the LR latent embedding directly? Furthermore, from past work (DiffBIR, PASD, SeeSR), we know that the role of the control branch is primarily to remove degradation and bring it closer to the HR distribution. However, the method proposed by the authors results in the distribution of the control branch outputs being closer to the distribution of LR latent embedding, which is puzzling.**
>
> A4: We appreciate your insightful question. Although we use LR latent embedding to constrain the control signal, this does not mean that the LR latent embedding can directly serve as the control signal. The LR latent embedding serves as input to the control branch and is inherently incompatible with the SD UNet. Through the control branch, the LR latent embedding is converted into the control signal that is adapted to the UNet, which is the purpose of diffusion adapters like ControlNet.
> During the conversion process of the LR latent embedding, as you mentioned, one role of the control branch is to make the control signal closer to the HR distribution, which is essentially a generative process. Since ControlNet is initialized using the parameters of the UNet, this process is also influenced by the generative prior of the UNet. However, during this process, the UNet's generative priors often distort the LR information, causing deviations in the generated output. In ClearSR, we similarly aim for the control branch to possess strong generative capabilities, but we correct the generative direction of the generative priors by adding additional constraints, resulting in output more consistent with the LR image.
>
> In Table 1 in the paper, it can be seen that our model exhibits stronger generative capabilities. Moreover, in Figure 2, we show that we successfully constrain the control signal in the latent space. These two results together demonstrate that our model achieves more "controllable" generation, and Figure 1 shows our model can generate results that are more consistent with the LR images.

---

> ### Comment · Reviewer_h8Y1 · 2024-11-22
>
> Thank you for your reply.
>
> In the response to Q2, I learned that the LSA setting is a key factor in balancing generative capability and fidelity. However, it still does not demonstrate the role of the detail preservation module in enhancing fidelity. The authors can remove the LSA setting to validate whether the proposed clearsr has an advantage over other methods in terms of fidelity.  By the way, from Fig. 3 in the PASD paper (https://www.ecva.net/papers/eccv_2024/papers_ECCV/papers/01705.pdf), the inference strategies proposed by PASD is also helpful for generative capabilities.
>
> In the response of Q3, how are the reconstruction results calculated? Are they computed between the output of VAE with LR as input  and GT?

---

> > ### Author Response · Authors · 2024-11-22
> >
> > Thank you for your reply. Our response is as follows:
> >
> > **Q5: In the response to Q2, I learned that the LSA setting is a key factor in balancing generative capability and fidelity. However, it still does not demonstrate the role of the detail preservation module in enhancing fidelity. The authors can remove the LSA setting to validate whether the proposed ClearSR has an advantage over other methods in terms of fidelity.**
> >
> > A5: Thank you for recognizing the role of our LSA. Regarding the Detail Preservation Module (DPM), we conduct an ablation study to demonstrate its contribution to fidelity. It can be observed that without DPM, the model's fidelity decreases significantly.
> >
> > | | PSNR $\uparrow$ | SSIM $\uparrow$ | NIQE $\downarrow$ | MANIQA $\uparrow$ |
> > |-----------|-----------|-----------|-----------|-----------|
> > |ClearSR|27.62|0.7483|6.6334|0.6222|
> > |ClearSR w/o DPM|26.88|0.7081|6.0081|0.6196|
> >
> > Next, we test the results of removing the LSA setting, and the quantitative comparisons with other methods on the DRealSR test set are as follows:
> >
> > | | PSNR $\uparrow$ | SSIM $\uparrow$ | NIQE $\downarrow$ | MANIQA $\uparrow$ |
> > |-----------|-----------|-----------|-----------|-----------|
> > |Real-ESRGAN|28.64|0.8053|6.6928|0.4907|
> > |StableSR|28.03|0.7536|6.5239|0.5601|
> > |PASD|27.36|0.7073|5.5474|0.6169|
> > |DiffBIR|26.71|0.6571|6.3124|0.5930|
> > |SeeSR|28.17|0.7691|6.3967|0.6042|
> > |ClearSR|28.22|0.7538|6.0867|0.6246|
> > |ClearSR w/o LSA |28.11|0.7419|6.5289|0.6226|
> >
> > It can be observed that ClearSR w/o LSA demonstrates an advantage over PASD and DiffBIR in terms of fidelity, performing similarly to SeeSR and StableSR. In terms of generative metrics, our model performs better than all these methods. Our model leverages the generative priors of Stable Diffusion to generate rich details, but generating details often comes at the cost of decreased fidelity, which explains its slightly lower performance compared to GAN-based methods (such as Real-ESRGAN). In this context, LSA serves as a useful tool. As mentioned in the response to Q2, ClearSR-a performs better than GAN-based methods in PSNR.
> >
> > Moreover, the choice of window size in the window-based cross-attention layers of the DPM might be the reason why our model without LSA does not outperform SeeSR and StableSR in terms of fidelity. It can be observed that increasing the window size leads to a decrease in fidelity, while decreasing it reduces the model's generative ability. To balance fidelity and generation, we selected 16 as the window size for ClearSR.
> >
> > | window size| PSNR $\uparrow$ | SSIM $\uparrow$ | NIQE $\downarrow$ | MANIQA $\uparrow$ |
> > |-----------|-----------|-----------|-----------|-----------|
> > |32|27.29|0.7294|6.5364|0.6333|
> > |16|27.62|0.7483|6.6334|0.6222|
> > |8|27.97|0.7619|6.9919|0.6090|
> >
> > **Q6: From Fig. 3 in the PASD paper (https://www.ecva.net/papers/eccv_2024/papers_ECCV/papers/01705.pdf), the inference strategies proposed by PASD is also helpful for generative capabilities.**
> >
> > A6: Thank you for your reminder. As shown in Figure 3 of the PASD paper, we can observe that as $\mathbf{\bar{\alpha}}_a$ increases, PSNR improves while QAlign decreases, indicating an increase in fidelity and a reduction in generative capability. According to the description in Section 3.3 of the PASD paper, when $\mathbf{\bar{\alpha}}_a$ is 0, it represents the original inference process. Increasing $\mathbf{\bar{\alpha}}_a$ means increasing the scale of adding LR latent. Based on this analysis, we can conclude that PASD's proposed ANS can only enhance fidelity by adjusting $\mathbf{\bar{\alpha}}_a$ but cannot improve the model's generative capability. If our explanation still confuses you, please feel free to submit a new response to us anytime.
> >
> > **Q7: In the response of Q3, how are the reconstruction results calculated? Are they computed between the output of VAE with LR as input and GT?**
> >
> > A7: Sorry for the confusion caused by our response. In the response to Q3, the reconstruction results are computed between the output of the VAE with GT as input and the GT.
> >
> > In this response, to better demonstrate the performance of VAE, we conduct additional tests on the DRealSR test set. We calculate the PSNR, SSIM, and LPIPS between the resized LR and GT images. We also calculate the PSNR, SSIM, and LPIPS between the outputs of the VAE with resized LR as inputs and the GT images. It can be seen that the outputs of VAE with resized LR as inputs show almost no decline in fidelity, and with PSNR and SSIM significantly outperforming existing methods (Real-ESRGAN, SeeSR, and ClearSR*).
> >
> > || PSNR | SSIM | LPIPS|
> > |-----------|-----------|-----------|-----------|
> > |LR| 30.57 | 0.8301 | 0.4608 |
> > |The outputs of VAE with resized LR as inputs| 30.53 | 0.8290 | 0.4602 |
> > |Real-ESRGAN| 28.64 | 0.8053 | 0.2847|
> > |SeeSR| 28.17 | 0.7691 | 0.3189|
> > |ClearSR*| 29.00 | 0.7781 | 0.3281 |

---

> > ### Author Response · Authors · 2024-12-01
> >
> > Dear Reviewer h8Y1,
> >
> > Thank you once again for your valuable feedback. We have carefully addressed your comments and have revised our paper accordingly. If you have any further questions, we would be eager to engage in further discussion with you.
> >
> > Additionally, we would like to take this opportunity to extend our warmest wishes for a joyful and restful Thanksgiving holiday to you and your team.
> >
> > Best regards,
> >
> > Authors of Submission 6007

---

### Meta-Review · Area_Chair_FyuV · 2024-12-16

**Metareview:**

This paper proposes ClearSR for image super-resolution. The main contribution of this work is the better use of the embeddings of the LR input image. Specifically, two modules, DPM and GSPM, are proposed to better encode information from the LR embedding, leading to enhanced output quality.

This paper is well motivated, highlighting the fidelity problem of existing super-resolution methods. In addition, the proposed method demonstrates decent performance in the evaluation datasets.

Despite the above strengths, the effectiveness of the proposed modules are not fully verified, some of which are shared by the reviewers. In particular,

1. The theoretical or intuitive explanation of the modules' effectiveness are not convincing. Why does attention help preserving details, and why are convolution blocks good at structure preservation? These claims are not carefully verified.

2. The comparison to `using only LR latent` is missing. This ablation is essential as it is a direct proof of the proposed modules leading to a better restoration. Since existing works have different settings and designs, the comparison with them could not lead to the conclusion that the propose model is effective.

3. The improvements over existing works are not conclusive. As also demonstrated by the authors, CFG, window size, text prompts, and other factors also lead to non-negligible quality differences. Given the inconclusive metrics (e.g., in Table 1), it is hard to claim that the proposed modules lead to positive effects. More ablations are needed to demonstrate the contributions in this work.

Based on the above considerations, the AC would recommend a rejection.

**Additional Comments On Reviewer Discussion:**

Reviewers generally question about the novelty and effectiveness of the proposed modules, and the complexity of the method. The authors address most of them with quantitative comparisons. The AC acknowledges the efforts of the authors and agrees on some of their explanations. However, the AC shares some of the reviewers' concerns about the effectiveness, as mentioned in the metareview. The authors are advised to take the reviewers' comments into consideration in the future version.

---

### Decision · Program_Chairs · 2025-01-22

Reject